

# Evaluating Simulations of Organic Aerosol Volatility and Degree of Oxygenation in Eastern China

Yu Li[1], Momei Qin[1*], Weiwei Hu[2], Bin Zhao[3], Ying Li[4], Havala O. T. Pye[5], Jingyi Li[1], Linghan Zeng[6], Song Guo[7], Min Hu[7], Jianlin Hu[1]

[1] Jiangsu Key Laboratory of Atmospheric Environment Monitoring and Pollution Control, Collaborative Innovation Center of Atmospheric Environment and Equipment Technology, School of Environmental Science and Engineering, Nanjing University of Information Science and Technology, Nanjing 210044, China

[2] State Key Laboratory of Organic Geochemistry, Guangzhou Institute of Geochemistry, Chinese Academy of Sciences, Guangzhou 510640, China

[3] State Key Laboratory of Regional Environment and Sustainability, School of Environment, Tsinghua University, Beijing 100084 China

[4] School of Environmental Science and Technology, Dalian University of Technology, Dalian 116024, China

[5] Gillings School of Global Public Health, University of North Carolina at Chapel Hill, Chapel Hill, North Carolina 27599, United States

[6] State Key Laboratory of Atmospheric Environment and Extreme Meteorology, Institute of Atmospheric Physics, Chinese Academy of Sciences, Beijing 100029, China

[7] State Key Laboratory of Regional Environment and Sustainability, International Joint Laboratory for Regional Pollution Control, Ministry of Education (IJRC), College of Environmental Sciences and Engineering, Peking University, Beijing 100871, China

*Corresponding to:* Momei Qin (momei.qin@nuist.edu.cn)

**Abstract.** Volatility and oxygen-to-carbon (O/C) molar ratios are critical properties of organic aerosols (OA), influencing their viscosity, hygroscopicity, and light absorption thereby resulting in impacts on air quality and climate. While atmospheric models often track these properties to simulate OA evolution, their performance remains insufficiently evaluated. This study assessed OA volatility and O/C simulations by comparing CMAQ model outputs using official AERO7i and community-contributed two-dimensional volatility basis set (2D-VBS) schemes, against two field measurements in eastern China. Apart from baseline modelling, two additional simulations using AERO7i incrementally incorporated low-volatility/semi-volatile/intermediate-volatility organic compound (L/S/IVOC) emissions and enhanced anthropogenic secondary organic aerosol (SOA) yields. An optimized 2D-VBS simulation further constrained O/C ratios of primary organic aerosol (POA)

emissions using observations. The results showed that OA mass concentrations were underestimated by 24% in 2D-VBS and 27-34% with updated AERO7i, likely due to underrepresented vehicular POA emissions and nighttime SOA formation. All simulations captured the substantial contribution of low-volatility products ($C^*$ <0.1 μg m$^{-3}$) but failed to reproduce the detailed volatility distributions within this range. Simulated O/C ratios were biased low in aged air masses (notably with 2D-VBS) and slightly overestimated in areas with more local emissions using updated AERO7i. Misrepresentations of OA volatility primarily led to biases in viscosity predictions, while the hygroscopicity parameter played a more important role. These findings highlight the need to better constrain OA volatility and O/C in models to improve projections of OA air quality and climate impacts.

## 1 Introduction

Organic aerosols (OA) are a major component of fine particulate matter (PM$_{2.5}$), accounting for 20–90% of its mass (Jimenez et al., 2009; Zhang et al., 2007), and play a critical role in global air quality, climate, and public health (Nault et al., 2021; Shrivastava et al., 2017; Wu et al., 2018). OA can be classified as either primary organic aerosol (POA), emitted directly from sources such as combustion, or secondary organic aerosol (SOA), formed through the oxidation of volatile organic compounds (VOCs) and semi-volatile and intermediate volatility organic compounds (S/IVOCs) (Hallquist et al., 2009; Heald and Kroll, 2020). The chemical aging of OA involves functionalization (incorporation of oxygen-containing groups), fragmentation, and oligomerization. These processes alter OA's composition and physicochemical properties, including volatility, oxidation state, viscosity, hygroscopicity, and light absorption (Chacon-Madrid and Donahue, 2011; Tritscher et al., 2011; Massoli et al., 2010; Rothfuss and Petters, 2017; Hems et al., 2021). Understanding these evolving properties is crucial for accurately predicting the impacts of OA on climate, air quality, and public health in atmospheric and climate models (Tsigaridis and Kanakidou, 2018).

Among the various properties of OA, volatility and the oxygen-to-carbon molar ratio (O/C, an indicator of the extent of oxygenation) are pivotal for constraining their atmospheric fate and impacts, and they are therefore important parameters in



chemical transport models (CTMs) (Donahue et al., 2013; Rao and Vejerano, 2018). The two-dimensional volatility basis set (2D-VBS), a widely adopted framework in OA modeling, tracks volatility and O/C to simulate chemical evolution (Donahue et al., 2011). Volatility primarily governs the partitioning of organic compounds between particulate and gas phases at

equilibrium, thereby influencing their atmospheric behavior (e.g., chemical aging, transport, and deposition) and overall OA concentrations (Donahue et al., 2014; Shiraiwa and Seinfeld, 2012). Li et al. (2020) recently parameterized the glass transition temperature ($T_g$, at which a phase transition between amorphous solid and semi-solid states occurs, and viscosity changes dramatically) as a function of volatility and O/C. This parametrization highlights the link between volatility and phase state (or viscosity), which influence the kinetics of gas-particle interactions, with implications for diffusion, partitioning, and

heterogeneous reaction rates (Zaveri et al., 2014; Reid et al., 2018; Marshall et al., 2018; Li and Shiraiwa, 2019). Furthermore, OA volatility affects cloud condensation nuclei (CCN) activity due to its connection with particle hygroscopicity and growth dynamics (Liu and Matsui, 2022; Zhang et al., 2023).

The O/C ratios of OA influence their viscosity and hygroscopicity, similar to the effects of volatility (Massoli et al., 2010;

Koop et al., 2011). The O/C ratios and volatility of OA components are largely coupled, with more volatile components typically exhibiting lower O/C ratios. Lower O/C ratios favor phase separation under specific relative humidity conditions (Pye et al., 2017). Conversely, elevated O/C ratios indicate increased hygroscopicity resulting in enhanced water uptake, thereby increasing their potential to act as cloud condensation nuclei and ice nuclei (Mahrt et al., 2022; Malek et al., 2023; Song et al., 2012; Tian et al., 2022). Furthermore, O/C ratios substantially impact the optical properties of OA (Xu et al., 2024).

During aging, bulk OA generally darkens with increasing O/C, while excessive oxidation at higher O/C levels diminishes light absorption (Jiang et al., 2022; Duan et al., 2024). These findings suggest that O/C could serve as a critical parameter in radiative forcing estimation in climate models.

A few prior studies have revealed significant uncertainties in OA volatility and O/C simulations with CTMs. For instance,

Saha et al. (2017) demonstrated that the VBS module in WRF-Chem significantly underestimated OA concentrations, and




failed to reproduce low-volatility OA components with effective saturation concentration ($C^*$) between $10^{-4}$ and $10^{-1}$ µg m$^{-3}$. This discrepancy could be partially explained by underestimated SOA formation due to wall losses of condensable vapors, and missing yields of low-volatility products with the parameters empirically derived from aerosol growth experiments. In contrast, parameters derived from dual thermodenuder (TD) measurements predicted 2–4× higher SOA yields, and produced more low-

volatility products under atmospherically relevant conditions (Saha and Grieshop, 2016). Additionally, the lack of S/IVOC emissions in the model likely contributed to biases in simulated OA volatility distributions (Xu et al., 2019). Regarding O/C simulations, Tsimpidi et al. (2018) introduced the ORACLE 2-D module into a global chemistry-climate model, which tended to overpredict OA O/C ratios in urban downwind areas. Overall, the model exhibited a 5–7% overestimation of O/C for OA and SOA, with the most pronounced positive biases in summer. The regional model CMAQ also overestimated OM/OC ratio

(closely related to O/C) of OA compared to observations in the southeastern US (Pye et al., 2017). However, accounting for interactions between OA and aerosol water, which enhance semi-volatile partitioning by increasing the available partitioning medium, could reduce model biases in OM/OC. The air quality model Polyphemus underestimated O/C in the northwestern Mediterranean region, even after implementing multi-generational oxidation processes, highlighting insufficient representation of aging processes in the model (Chrit et al., 2018). While model performance varies across different regions and models

globally, simulations of OA volatility and O/C ratios in the polluted atmospheres of China remain insufficiently explored.

In this study, we evaluated the performance of the Community Multiscale Air Quality (CMAQ) model in simulating OA mass concentration, volatility distribution and O/C ratio, by comparing with the observations at two sites in eastern China. The evaluation focused on two OA schemes that differed in their level of complexity regarding the representation of volatility and

O/C. Additionally, two modifications of the default CMAQ scheme were assessed to investigate the impacts of S/IVOC emissions and updated aromatic (i.e., benzene, toluene and xylene) and polycyclic aromatic hydrocarbon (PAH) SOA yields. The assessment also explored how uncertainties in OA volatility and O/C propagated to $T_g$ and viscosity predictions. These efforts aimed to provide valuable insights into improving OA simulations (in terms of both mass concentrations and properties), and their implications for phase state and viscosity modeling in CTMs.



## 2 Methods and Data

### 2.1 Model configuration

In this study, two versions of the CMAQ model (https://epa.gov/cmaq), the official v5.3.2 (Appel et al., 2021) (available from https://doi.org/10.5281/zenodo.4081737) and v5.4 with a community contribution (available from https://doi.org/10.5281/zenodo.7218076), were used to perform nested simulations with horizontal resolutions of 36 km and 12 km. The outer domain encompassed most of China, while the inner domain focused on eastern China (Fig. S1). The simulation period covered March 17–April 21 and September 29–November 21, 2018, aligning with the campaigns conducted in Dongying (DY) and Guangzhou (GZ). Meteorological fields were generated using the Weather Research and Forecasting (WRF) model version 4.2.1. Anthropogenic emissions for China were represented using the 2018 high-resolution (0.25°×0.25°) Multi-resolution Emission Inventory for China (MEIC) v1.4 (http://www.meicmodel.org), while the Regional Emission Inventory in Asia (REAS) v3.2.1 (https://www.nies.go.jp/REAS/) was used for the rest of Asia. Emissions from open burning were obtained from the Fire Inventory from NCAR (FINN) v1.5 (https://www2.acom.ucar.edu/modeling/finn-fire-inventory-ncar), and biogenic emissions were calculated using the Model of Emissions of Gases and Aerosols from Nature (MEGAN) v2.1 (Guenther et al., 2012). The simulations utilized the SAPRC07tic gas-phase mechanism within CMAQ (Xie et al., 2013). The capabilities of WRF and CMAQ in simulating meteorological factors and major pollutants ($NO_2$, $O_3$ and $PM_{2.5}$) were evaluated (Tables S1-S2).

### 2.2 OA representations

All the simulations modeled SOA formation from isoprene, glyoxal, and methylglyoxal, as well as $NO_3$-initiated oxidation of monoterpenes, using consistent parameterizations from CMAQ's official AERO7i module (see Fig. S2). Specifically, isoprene SOA formed via the aqueous uptake of isoprene epoxydiols (IEPOX)/methacrylic acid epoxide (MAE), and monoterpene-derived organic nitrates (ONs) via $NO_3$ oxidation were explicitly represented (Pye et al., 2013; Pye et al., 2015). Semi-volatile isoprene SOA, including contributions from non-aqueous pathways, was parameterized with the two-product model (Carlton et al., 2010), which utilized mass yields of two products ($\alpha_i$, $i$=1, 2) and their effective saturation concentrations ($C_i^*$, $i$=1,2) to simulate aerosol growth observed in experiments (Pankow, 1994). All the simulations accounted for multiphase SOA formation



from glyoxal and methylglyoxal both in cloud (Carlton et al., 2008) and on wet aerosol surfaces (Pye et al., 2015).


Five simulations were conducted to evaluate model performance regarding OA mass concentrations, volatility distributions, and O/C ratios against observational data (Table 1). All simulations retain the same AERO7i heterogeneous chemistry and select other systems (see above) and focus on different representations of key volatility-based systems (Table 1). The simulation using the standard AERO7i treatment including SOA from anthropogenic precursors (e.g., aromatics, long-chain alkanes) (Qin

et al., 2021), monoterpene oxidation via $O_3$/OH oxidation (Xu et al., 2018), and sesquiterpenes (Carlton et al., 2010) is referred to as a 1-D VBS for simplicity. In another simulation, the 2D-VBS framework (Zhao et al., 2015; Zhao et al., 2016; Chang et al., 2022) was used for anthropogenic VOCs, monoterpenes (excluding $NO_3$ oxidation pathways), and sesquiterpenes. Additionally, in 2D-VBS, the oxidation of POA (or L/S/IVOCs) was represented differently compared to the 1D-VBS simulations (Murphy et al., 2017).

**Table 1: Description of the five simulations conducted in this study.**

| Case | OA modeling |
|---|---|
| 1D-VBS | Default CMAQv5.3 AERO7i configuration |
| 1D-VBS_E | 1D-VBS with added emissions of L/S/IVOCs, with SOA formation from IVOC oxidation and an updated volatility distribution of semi-volatile POA |
| 1D-VBS_EY | 1D-VBS_E with updated SOA yields for aromatic and PAH precursors, accounting for vapor wall loss effects and autoxidation pathways |
| 2D-VBS | CMAQv5.4 with the 2D-VBS community contribution. Emission inputs closely align with the 1D-VBS_E simulation |
| 2D-VBS_A | 2D-VBS with modified O/C ratio distribution of POA, optimized based on observational constraints |

The 1D-VBS simulation reflects the default CMAQ configuration without modifications to emissions or chemistry. The 1D-



VBS_E simulation incorporated additional IVOC emissions, and updated emissions of S/LVOCs (historically classified as POA), based on MEICv1.4 (Supplementary Note 1). L/SVOC and IVOC emissions in each volatility bin were scaled from

POA and total VOC emissions, respectively, using source-specific scaling factors (Table S3). Since the lowest volatility bin for emissions in the standard 1D-VBS parameterization was set to $C^*=10^{-1}$ μg m$^{-3}$, the difference between original POA emissions and L/SVOC emissions within the model-resolved volatility range ($-1\leq\log C^*$(μg m$^{-3}$)$\leq2$) was reclassified as non-volatile ($C^*=10^{-10}$ μg m$^{-3}$, see Fig. S3) and treated with heterogeneous aging chemistry (Simon and Bhave, 2012). This adjustment conserved POA mass while addressing the model's limitations in volatility coverage. The estimated nation-wide

L/SVOC and IVOC emissions, which were 3.18 Tg yr$^{-1}$ and 6.68 Tg yr$^{-1}$, respectively, were higher than those reported by Zheng et al. (2023) but agreed well with Chen et al. (2024a) in magnitude (Table S4, Figs. S4-S5). However, the source contributions and volatility distributions of L/S/IVOCs were slightly different. The aging of L/SVOCs followed the POA treatment established in prior studies (Donahue et al., 2012; Murphy et al., 2017), where a fraction of L/SVOCs was oxidized to form SOA in the gas phase and the volatility distribution in continually updated. The IVOC-derived SOA formation adopted

the parameterization of Lu et al. (2020). In the 1D-VBS_EY simulation, SOA yields for benzene, toluene, xylene, and naphthalene oxidation (Table S5) were updated to account for vapor wall losses and formation of highly oxygenated organic molecules (HOMs) via autoxidation (Bilsback et al., 2023).

The 2D-VBS scheme (Zhao et al., 2015; Zhao et al., 2016; Chang et al., 2022), was implemented into CMAQv5.4 and

evaluated using emission inputs largely consistent with the 1D-VBS_E simulation. The major differences in emissions included (1) reallocating non-volatile POA to the LVOC species with $C^*$ of $10^{-2}$ μg m$^{-3}$ (i.e., the least volatile primary emission category in 2D-VBS); and (2) mapping SVOC emissions with $C^*$ of $10^{-1}$ μg m$^{-3}$ to the $C^*=10^{-2}$ μg m$^{-3}$ bin as well (due to the absence of a corresponding volatility bin in the 2D-VBS scheme). L/SVOC aging products were classified as SOA. Initial O/C ratio distributions for L/S/IVOC emissions across volatility bins followed 2D-VBS default profiles. However, these default

assumptions led to underestimated POA O/C ratios (see Section 3.2). Therefore, we conducted the 2D-VBS_A simulation, wherein sector-specific O/C ratios for POA emissions from gasoline/diesel vehicles, non-road mobile sources, power plants



and industrial sources were constrained using observational averages in eastern China (Table S6).

**2.3 Volatility distribution and O/C ratio modeling**

The volatility (expressed in terms of $C^*$) and O/C ratios for OA surrogate species across all the simulations were summarized

in Tables S7-S8. Volatilities for all species, except those in VBS-based parameterizations, were aligned with surrogate data in chamber experiments. Non-volatile POA and SOA formed via pathways such as aromatic oxidation under low-$NO_x$ conditions, aqueous-phase uptake of IEPOX/MAE/glyoxal/methylglyoxal, ON hydrolysis and particle-phase oligomerization were assigned a fixed $C^*$ of $10^{-10}$ µg m$^{-3}$ for simplicity. The volatility of primary L/S/IVOCs in the condensed phase and aged OA, was described in detail by Murphy et al. (2017). In the 1D -VBS_EY simulation, a new SOA species with $C^*$=0.1 µg m$^{-3}$

(unrepresented in the base 1D-VBS in CMAQ) was explicitly incorporated. The volatility distribution of total SOA (POA) was then derived using simulated concentrations of individual SOA (POA) species.

The overall O/C ratios for SOA, POA, and OA were calculated using the mole-weighted averages of the O/C values for individual species. The O/C ratios of OA species were calculated based on mass-based OM/OC ratios provided with the CMAQ

code. The OM/OC ratios for SOA were primarily derived from chamber experiments (Pye et al., 2017; Carlton et al., 2010; Xu et al., 2018). For example, SOA species with relatively well-known structures (e.g., IEPOX-/MAE-derived SOA, isoprene dinitrates and monoterpene nitrates), along with seven VBS bins representing monoterpene SOA formed via $O_3$ and OH oxidation, adopted OM/OC values and other molecular properties from surrogate compounds identified in chamber studies. For other SOA species, once their $C^*$ values were determined (as described above) and the number of carbons ($n$C) was

assumed, the OM/OC ratios were either inferred based on plausible structures (Pankow et al., 2015) or estimated using the relationship between volatility, $n$C and the number of oxygens ($n$O) used in 2D-VBS (Pye et al., 2017). Additionally, OM/OC ratios for primary L/S/IVOCs and their aging products were constrained by laboratory work and filed observations as documented in previous studies (Murphy et al., 2017; Lu et al., 2020). The non-volatile POA emissions were assumed to have an OM/OC ratio of 1.6 (Turpin and And Lim, 2001), which was equivalent to O/C of 0.35.




In the base 2D-VBS, the O/C distributions for L/S/IVOC emissions in each volatility bin followed the default settings in the community-contributed 2D VBS. These O/C ratios were primarily based on Chang et al. (2022), which used emission test data for specific sectors when available and otherwise adopted values from prior studies. However, in the 2D VBS, the O/C ratios for emissions were capped at 0.4. For SOA products overlapping with AERO7i, such as isoprene SOA, the O/C ratios were set to match those in AERO7i. However, for other SOA surrogate species, the O/C ratios were determined either by adjusting the O:C distribution of first-generation products to align with experimental data (OH/O₃-intitiated monoterpene and sesquiterpene SOA), or based on explicit chemical mechanisms for initial oxidation (anthropogenic SOA), followed by aging through functionalization and fragmentation within 2D-VBS (Zhao et al., 2015). As a result, the O/C ratios for OH/O₃-intitiated monoterpene and sesquiterpene SOA in 2D-VBS generally ranged from 0 to 1.0 in 0.1 increments. In contrast, anthropogenic SOA (ASOA) exhibited a broader range, with a maximum O/C ratio of 2.0, to account for the high degree of oxygenation observed in toluene SOA in chamber experiments.

**2.4 Predictions of Tg and viscosity**

Previous measurements have demonstrated a close relationship between volatility and viscosity. For instance, Champion et al. (2019) found that SOA with higher fractions of EL/LVOCs showed increased viscosity. Similarly, an inverse correlation between $T_g$ and vapor pressure was observed for isoprene SOA components (Zhang et al., 2019). Although the dependence of $T_g$ on atomic O/C ratios is generally weaker than on vapor pressure (Koop et al., 2011), strong correlations between $T_g$ and O/C ratios have been observed for oxidation products formed from specific precursors, such as α-pinene (Dette et al., 2014), n-heptadecane, and naphthalene (Saukko et al., 2012). Therefore, the uncertainties in volatility and O/C ratios may impact $T_g$ and viscosity predictions when using the parameterizations that relate $T_g$ and viscosity to volatility and O/C ratios (Li et al., 2020; Zhang et al., 2019).

Here we calculated the $T_g$ of individual OA surrogate species $i$ ($T_{g,i}$) using Eqs. (1), (2) (Li et al., 2020) or (3)(Zhang et al.,



2019):

$$T_{g,i} = 288.70 - 15.33 \times log_{10}(C_i^0) - 0.33 \times [log_{10}(C_i^0)]^2 \qquad (1)$$

$$T_{g,i} = 289.10 - 16.5 \times log_{10}(C_i^0) - 0.29 \times [log_{10}(C_i^0)]^2 + 3.23 \times log_{10}(C_i^0) \times (O/C) \qquad (2)$$

$$T_{g,i} = 480.07 - \frac{54395}{\left(log_{10}\left(\frac{RT}{M_i}C_i^0\right) - 7.7929\right)^2 + 116.49} \qquad (3)$$

Where Eq. (1) relies solely on volatility ($C_i^0$ represents the saturation concentration at 298K, and is equal to $C_i^*$ assuming ideal thermodynamic mixing), and was developed for coupling into the 1-D VBS framework. Equation (2) incorporates O/C as an additional factor, which is used in 2D-VBS. Both equations yield similar predictions, particularly for compounds with low O/C ratios (Li et al., 2020). Equation (3) is a semi-empirical formula derived from the measurements of isoprene SOA components, which relates $T_g$ to volatility and includes molar mass $M_i$ (Zhang et al., 2019). The overall $T_g$ of OA mixtures under dry conditions ($T_{g,org}$) can be calculated using the Gordon-Taylor equation (see Supplementary Note 2).

OA viscosity depends on aerosol water content, since water significantly influences the phase state of aerosols (Koop et al., 2011). Under humid conditions, the $T_g$ of organic-water mixtures ($T_{g,\omega org}$) is calculated based on the mass fraction of organics ($\omega_{org}$) in the mixtures, along with the $T_g$ values of pure water (136 K) and dry OA (using Eqs (1), (2) or (3)). Notably, $\omega_{org}$ varies with ambient relative humidity (RH) and the effective hygroscopicity parameter of OA ($\kappa_{org}$). Further details can be found in previous studies (Derieux et al., 2018; Shiraiwa et al., 2017) and Supplementary Note 2. The temperature-dependent viscosity ($\eta$) can be estimated using the modified Vogel-Tammann-Fulcher (VTF) equation, as given in Eq. (4) (Angell, 1991), when the ambient temperature $T$ is at or above $T_{g,\omega org}$; otherwise $\eta$ is fixed at $10^{12}$ Pa s.

$$\eta = \eta_\infty e^{\frac{T_0 D}{T - T_0}} \qquad (4)$$

where $\eta_\infty$ and $D$ denote the viscosity at infinite temperature and the fragility parameter, with the values of $10^{-5}$ Pa s and 10, respectively (Derieux et al., 2018). $T_0$ represents the Vogel temperature, and can be estimated using Eq. (5) (Angell, 1991; Derieux et al., 2018):



$$T_0 = \frac{39.17 T_{g,\omega org}}{D + 39.17}$$
(5)

### 2.5 Measurement Data

Two sets of observational data from field campaigns were used to evaluate the simulations. One campaign was conducted in

the spring (March 17–April 21) of 2018 in DY, located downwind of the North China Plain (NCP) (Feng et al., 2023). The

observational data include the mass concentration and elemental ratios (e.g., O/C, H/C) of OA, measured using a high-

resolution time-of-flight aerosol mass spectrometer (HR-AMS). The volatility distribution of OA (expressed as VBS) was

estimated using an empirical method based on data from a thermodenuder (TD) combined with AMS. The sources of OA

components were characterized using the positive matrix factorization (PMF) method, which resolved OA factors, including

hydrocarbon-like OA (HOA), biomass-burning OA (BBOA), transported OOA and background OOA. The other campaign,

led by the same team, was conducted in the autumn (September 29–November 21) of 2018 in urban GZ (Chen et al., 2021).

The resolved OA factors in GZ identified three POA groups (cooking OA (COA), HOA, and nitrogen-containing OA (NOA))

along with two types of SOA (low-volatility oxygenated OA, LV-OOA, and semi-volatile oxygenated OA, SV-OOA).

### 3 Results

### 3.1 Evaluation of mass concentration

Figure 1 compared the observed and simulated diurnal variations of OA and its components, POA and SOA, in DY and GZ,

respectively. The simulation that updated O/C ratios of POA in 2D-VBS (i.e., 2D-VBS_A) didn't meaningfully affect the

predicted mass concentration or volatility distribution, and is therefore not discussed in Section 3.1 and 3.2. The observations

showed that SOA dominated OA at both locations (contributing 72% and 64%, respectively), while the model underestimated

this contribution (59%-67%) in DY with the exception of 2D-VBS (82%), and overestimated (72%-84%) it in GZ (Table S9).

In general, the 2D-VBS case predicted higher contributions of SOA to total OA (>80%) than in the 1D-VBS simulations.





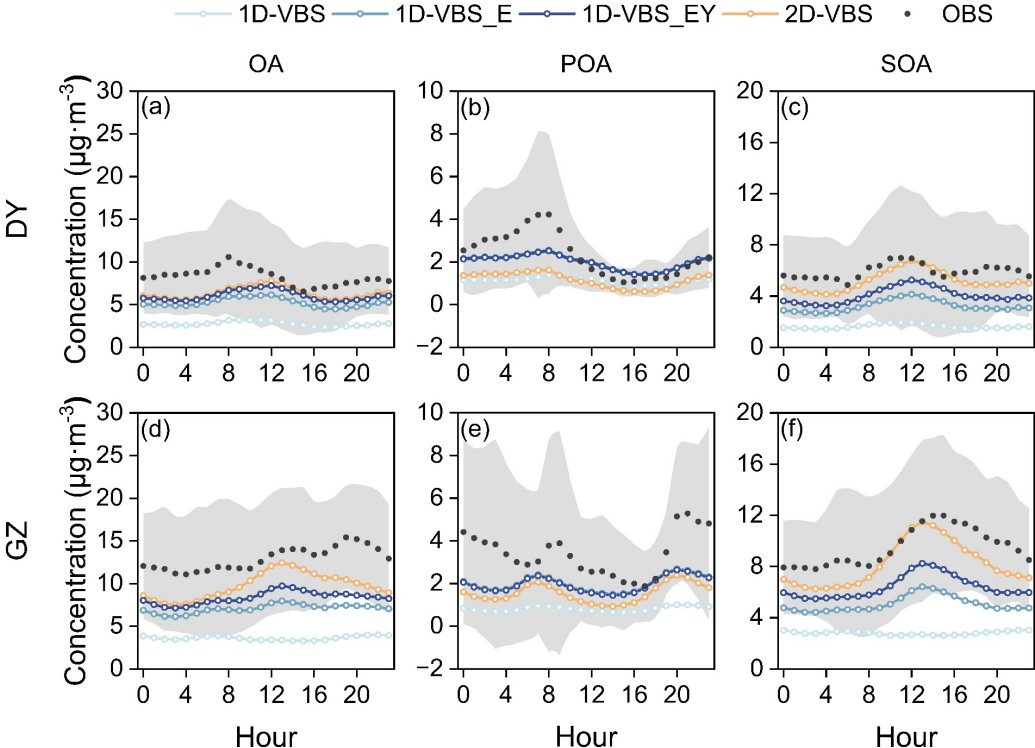

**Figure 1: Simulated diurnal variations of OA, POA, and SOA with different SOA schemes compared to the observations in DY (a-c) and GZ (d-f). The light grey shading indicates the standard deviations ($\pm 1\sigma$) from the mean concentration in the observations.**

The base 1D-VBS significantly underestimated SOA production, with NMBs of −72% in DY and −71% in GZ (Table S10). Modifications of L/S/IVOC emissions and their contributions to SOA in the 1D-VBS_E case resulted in approximately a two-fold enhancement in SOA mass concentration. This finding was consistent with earlier studies showing that 64% to 100% of observed SOA in GZ can be explained when both conventional precursors and S/IVOCs are considered (Hu et al., 2022), and that simulated S/IVOC contributions to SOA exceeded 50% across most of China (Li et al., 2021a; Miao et al., 2021). The 1D-VBS_EY case substantially increased SOA production from aromatics and PAHs with updated yields (Fig. S6), and the diurnal variation was similar to that in the 2D-VBS simulation in DY. In GZ, the notable daytime differences in SOA between the 2D-VBS and 1D-VBS_EY cases were driven by variations in ASOA representations (Fig. S7). Mass increases in ASOA as a result

of gas-phase aging were likely overestimated with the 4-km grid spacing in the 1D-VBS_EY simulation (Bilsback et al., 2023).

While both simulations significantly improved SOA predictions, they still underestimated SOA concentrations compared to

observations by 11-14% (2D-VBS) and 30-33% (1D-VBS_EY), respectively. Diagnosing the causes of these underestimations

remains challenging due to the lack of chemically resolved SOA measurements. We infer that the nighttime low biases in both

cases were likely attributed to insufficient formation of organic nitrates in the presence of $NO_3$ radicals. This might result from

missing anthropogenic terpene and phenolic emissions from sources such as biomass burning and volatile chemical products

(Wang et al., 2022; Coggon et al., 2021; Xie et al., 2025; Liu et al., 2024). The underestimated nocturnal SOA formation could

also be attributed to the underprediction of aqueous-phase formation pathways, which are enhanced under high RH conditions

at night (Wang et al., 2019; Gu et al., 2022).

Simulated POA also exhibited underestimation at both locations. In the comparison, BBOA and COA were excluded from the

observations, due to the complexity and considerable uncertainties associated with cooking and biomass burning (particularly

open burning) emission estimation (Li et al., 2023; Zhou et al., 2017). Therefore, the POA underestimation was most likely

attributable to uncertainties in mobile emissions. Observed POA concentrations at both sites showed peaks during traffic rush

hours, which were not well captured by the simulations. At the DY site, the resolved HOA in observations was partially aged,

as noted in Feng et al. (2023) and included some SOA transported over long distances, which may also explain the POA

underestimation. At the GZ site, which is more influenced by local emissions, the simulated peak concentration exhibited an

earlier shift compared to the observed peak, suggesting that the diurnal patterns of POA emissions may differ from those

specified in the model. Additionally, the POA volatility distributions (see Methods) and aging schemes in the 1D-VBS and

2D-VBS simulations were different: (1) The 1D-VBS involved heterogeneous aging of non-volatile POA. (2) In 2D-VBS,

POA aging led to its complete transformation into SOA, whereas in the 1D-VBS, some oxidation products remained as POA.

These differences led to distinct POA levels between the two cases.


Overall, total mass concentrations of OA were underestimated. The 1D-VBS_EY case, which had the smallest negative biases



among the 1D-VBS simulations, underestimated OA by approximately 27% in DY and 34% in GZ. These discrepancies were dominated by SOA underestimation at both sites, with POA biases in the 1D-VBS negligible in GZ (Fig. S8). The 2D-VBS simulation demonstrated better performance in OA predictions than 1D-VBS_EY, with NMBs of −24% in both DY and GZ, primarily due to higher predicted SOA mass concentrations.

**3.2 Volatility simulations**

The mass distributions of OA, POA, and SOA across different volatility bins (i.e., volatility distribution) were examined. This study primarily focused on compounds with $C^*$ equal or below $10^{-1}$ µg m$^{-3}$ ($\log C^* \leq -1$, referred to as low-volatility OA hereafter), and on SVOCs (i.e., $-1 < \log C^* \leq 2$), which exist in both gas and condensed phases. Despite variations in OA mass concentrations across different cases, CMAQ generally reproduced the observed volatility distributions of OA, SOA, and POA in all the simulations (Fig. 2). Specifically, the observations indicated that more than 70% of OA was low-volatility OA, consistent across the simulations. The base 1D-VBS significantly underestimated low-volatility OA concentrations, leading to underestimation in total OA mass (Fig. S9). The 1D-VBS_E simulation, which included L/S/IVOC emissions, increased low-volatility SOA formation by 1.1 µg m$^{-3}$ in DY and 1.5 µg m$^{-3}$ in GZ, although the contributions of low-volatility SOA to total SOA decreased. It also replaced the default volatility distribution of POA in CMAQv5.3 with source-specific gas/particle partitioning, resulting in a significant amount of non-volatile POA (Fig. S10). Collectively, these adjustments led to slightly higher fractions of low-volatility OA with the inclusion of L/S/IVOC emissions, as proposed by Xu et al. (2019), albeit for a different reason (mainly due to changes in POA rather than SOA volatility distributions). The updated SOA yields for aromatics and PAHs had minor effects on POA but increased SOA mass concentrations across three of the four volatility bins in Fig. S9, slightly changing the SOA volatility distributions.



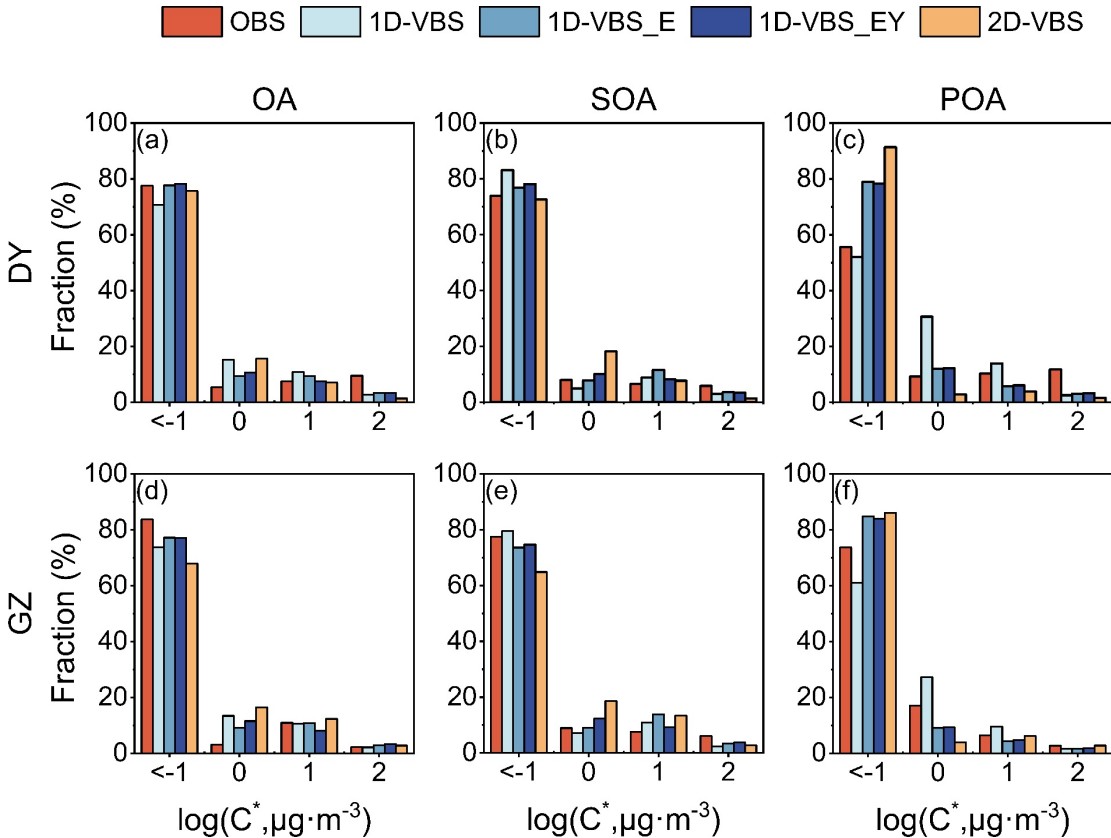

**Figure 2: Simulated volatility distributions of OA, POA, and SOA across volatility bins with $C^*$ up to $10^2$ µg m$^{-3}$, compared to the observations in DY (a-c) and GZ (d - f). The compounds with $C^*$ equal or below $10^{-1}$ µg m$^{-3}$ were grouped into a single volatility bin, and referred to as low-volatility OA in this study.**


The simulated OA volatility distributions, particularly for POA, differed between the 2D-VBS and 1D-VBS_EY simulations. The observations presented a higher fraction of low-volatility POA in GZ than in DY. The spatial variation in POA volatility was captured by the 1D-VBS_EY case. However, both 1D-VBS_EY and 2D-VBS overestimated the contributions of low-volatility POA. In 1D-VBS_EY, non-volatile POA was substantial in mass, and increased with aging (Fig. 3); semi-volatile

POA underwent aging, with a portion remaining as POA (rather than fully converting into SOA as in 2D-VBS) but becoming more volatile. This ultimately led to a higher low-volatility POA mass and a lower contribution compared to 2D-VBS. The



SOA volatility distributions in the 1D-VBS_EY and 2D-VBS cases both agreed well with the observations, which were similar

at the two locations. However, the 2D-VBS simulation predicted a lower fraction of low-volatility BSOA in GZ compared to

the 1D-VBS simulations (over 73% vs. less than 86%), primarily due to differences in the treatment of monoterpene-derived

SOA through O₃/OH oxidation pathways. Additionally, the contribution of low-volatility ASOA was also lower in the 2D-VBS

simulation (over 62% vs. over 69%; see Fig. S11).

**Figure 3: Simulated mass concentrations of OA, POA, and SOA across volatility bins with C\* ranging from $10^{-10}$ to $10^{2}$ μg m⁻³, compared to observations in DY (a-c) and GZ (d-f).**




It is important to note that LVOCs ($-4<\log C^*$ (μg m$^{-3}$) $\leq-1$) and ELVOCs (extremely low-volatility VOCs with $\log C^*$ (μg m$^{-3}$) $\leq-4$) were not explicitly resolved in this study (Fig. 3). In all simulations, SOA formation was dominated by species within $C^*$ bins ranging from $10^{-2}$ to $10^4$ μg m$^{-3}$. In addition, certain assumed "non-volatile" products (IEPOX/MAE-derived SOA, oligomers, etc.) were arbitrarily assigned to the volatility bin of $C^*$=$10^{-10}$ μg m$^{-3}$ (see Methods). The abundance of mass in the

$C^*$=$10^{-2}$ μg m$^{-3}$ bin was particularly high for the 2D-VBS and largely due to aging. The absence of predicted SOA between the $C^*$ bins at $10^{-10}$ μg m$^{-3}$ and $C^*$=$10^{-2}$ μg m$^{-3}$ was in contrary to the observed more uniform distribution across $C^*$ bins spanning from $10^{-9}$ to $10^{-1}$ μg m$^{-3}$. The 1D-VBS_EY case included formation of mass for the volatility bin at $C^*$=$10^{-6}$ μg m$^{-3}$ for HOMs formed from aromatics and PAHs under low-NO$_x$ conditions, but the predicted mass concentration was negligible and biased low relative to the observations. Major ELVOC formation pathways, including autoxidation and bimolecular peroxy radical

reactions for monoterpenes, have been incorporated into the Community Regional Atmospheric Chemistry Multiphase Mechanism (CRACMM) (Pye et al., 2023). However, the volatility distribution of the resulting products remained insufficiently resolved. Furthermore, anthropogenic ELVOC formation is not yet well understood (Shrivastava et al., 2024; Yin et al., 2024). As ambient observations consistently reported significant amounts of LVOCs and ELVOCs (Chen et al., 2024b; Huang et al., 2024), future work should refine and expand the representation of autoxidation and other chemical

processes contributing to LVOC and ELVOC formation from both anthropogenic and biogenic precursors. These improvements will enhance model accuracy in predicting SOA volatility and new particle formation.

**3.3 Oxygen to carbon ratio simulations**

Both observed and simulated OA showed higher O/C ratios during the day and lower ratios at night (Fig. S12), aligning with the temporal patterns of SOA mass concentrations. The higher O/C ratios in the afternoon were linked to elevated oxidant

levels, which facilitated SOA formation (typically characterized by higher O/C ratios compared to POA, see Table 2) and the photochemical aging of OA. In contrast, during the night and early morning, the increased contributions of fresh POA led to lower O/C ratios.



**Table 2: Simulated and observed O/C ratios of OA, POA and SOA in DY and GZ.**

| Site | Case | OA | | POA | | SOA | |
|------|------|-----|-----|-----|-----|-----|-----|
| | | SIM | OBS | SIM | OBS | SIM | OBS |
| DY | 1D-VBS | 0.52 | | 0.15 | | 0.81 | |
| | 1D-VBS_E | 0.56 | | 0.29 | | 0.77 | |
| | 1D-VBS_EY | 0.61 | 0.83 | 0.29 | 0.55 | 0.83 | 1.04 |
| | 2D-VBS | 0.49 | | 0.26 | | 0.57 | |
| | 2D-VBS_A | 0.50 | | 0.29 | | 0.57 | |
| GZ | 1D-VBS | 0.58 | | 0.15 | | 0.72 | |
| | 1D-VBS_E | 0.59 | | 0.33 | | 0.71 | |
| | 1D-VBS_EY | 0.66 | 0.59 | 0.33 | 0.25 | 0.78 | 0.82 |
| | 2D-VBS | 0.51 | | 0.20 | | 0.58 | |
| | 2D-VBS_A | 0.52 | | 0.25 | | 0.58 | |


Most simulations underestimated the O/C ratios of OA, with a more pronounced underestimation at the DY site, where both POA and SOA O/C ratios were lower than observed (Table 2). In DY, OA underwent prolonged aging during transport, resulting in a higher O/C ratio (0.84) than the national average (0.3–0.65) (Feng et al., 2023). Although the 1D-VBS_EY case predicted the highest O/C ratio of OA (due to potentially overestimated multi-generational oxidation as discussed in Section 3.1), it was

still lower than the observed value (0.61 vs. 0.83). This discrepancy suggests that accurately simulating the evolution of OA O/C ratios with aging remains a significant challenge for current CTMs (despite satisfactory mass simulations). The observed POA contained aged HOA, which, although essentially SOA, could not be separated from HOA in DY (Feng et al., 2023), partly explaining the underestimation of POA O/C ratios. In the 1D-VBS_E case, the inclusion of L/S/IVOC emissions increased the O/C ratio of POA but decreased that of SOA, consistent with the impacts on volatility, i.e., increased (decreased)

the contribution of low-volatility POA (SOA) (see Fig. 2 and Fig. S10). The 1D-VBS_EY case promoted SOA formation at



$C^*$=0.1 μg m$^{-3}$ through the newly added volatility bin for ASOA, which was highly oxygenated (O/C=2.53, see Table S7), thereby increasing the O/C ratios of SOA. However, the impacts of HOMs were negligible due to their minimum contribution to mass.

At the GZ site, the overestimation of POA O/C ratios in certain cases (i.e., 1D-VBS_E and 1D-VBS_EY) partially offset the underestimation of SOA O/C ratios. As a result, the O/C ratio of total OA in 1D-VBS_E was closest to the observed value (both were 0.59), while the 1D-VBS_EY case overpredicted the O/C ratio (0.66 vs. 0.59). The more oxygenated POA in both simulations than observed could be attributed to overestimations in non-volatile POA mass concentrations. Additionally, COA, with an observed O/C ratio of 0.19 in GZ, were likely underestimated in the simulations given uncertainties in cooking

emissions. If COA were well represented, it could slightly lower the simulated POA O/C ratios and reduce the positive biases in the 1D-VBS_E and 1D-VBS_EY simulations in GZ.

The O/C ratios predicted by the 2D-VBS case were lower than those in other simulations, primarily due to the underestimation in SOA O/C ratios. In particular, the distribution of ASOA O/C ratios differed significantly from those in the 1D-VBS

simulations (see Fig. S13), highlighting the necessity for improved representation of ASOA O/C ratios in future work. Simulated POA O/C ratios were mainly influenced by how emissions were specified in the model. As the default settings in 2D-VBS underestimated POA O/C ratios, an additional simulation, 2D-VBS_A, was conducted with updated O/C ratios for emissions, constrained by those of POA factors in observations from prior studies (Table S6 and Fig. S14). For instance, in the default configuration, mobile sources were assumed to emit primarily hydrocarbons and low-oxygenated compounds, with

over 90 % of emissions distributed in the O/C bins of 0 and 0.1. The optimized parameters in the 2D-VBS_A case reflected a higher degree of oxygenation, with more than 60 % of emissions allocated to the O/C bins of 0.2 and 0.4. As a result, the POA O/C increased from 0.26 (DY) and 0.20 (GZ) in 2D-VBS to 0.29 and 0.25 in 2D-VBS_A, respectively, and showed good agreement with the observation in GZ. However, the persistent low bias in POA O/C ratios in DY was likely attributable to uncertainties in emissions from sources with higher O/C, such as biomass burning (with an observed O/C ratio of 0.37 for





BBOA) in the simulations, as well as potential overestimation of observed POA O/C ratios due to the influence of aged HOA.

Apart from underestimations in OA O/C ratios in most cases, the simulations struggled to capture the spatiotemporal variability

on an hourly basis (Fig. 4). The observations revealed significant differences in the O/C ratios between DY and GZ, along with

considerable temporal variability, particularly at the GZ site with abundant fresh emissions. In the observations, the

interquartile range (IQR) of hourly O/C ratios was 0.77-0.89 in DY and 0.52-0.68 in GZ. However, the simulated O/C ratios

of OA showed less variability than observed in GZ, with narrower IQRs and Standard Deviation Ratios (SDR) lower than 1.0.

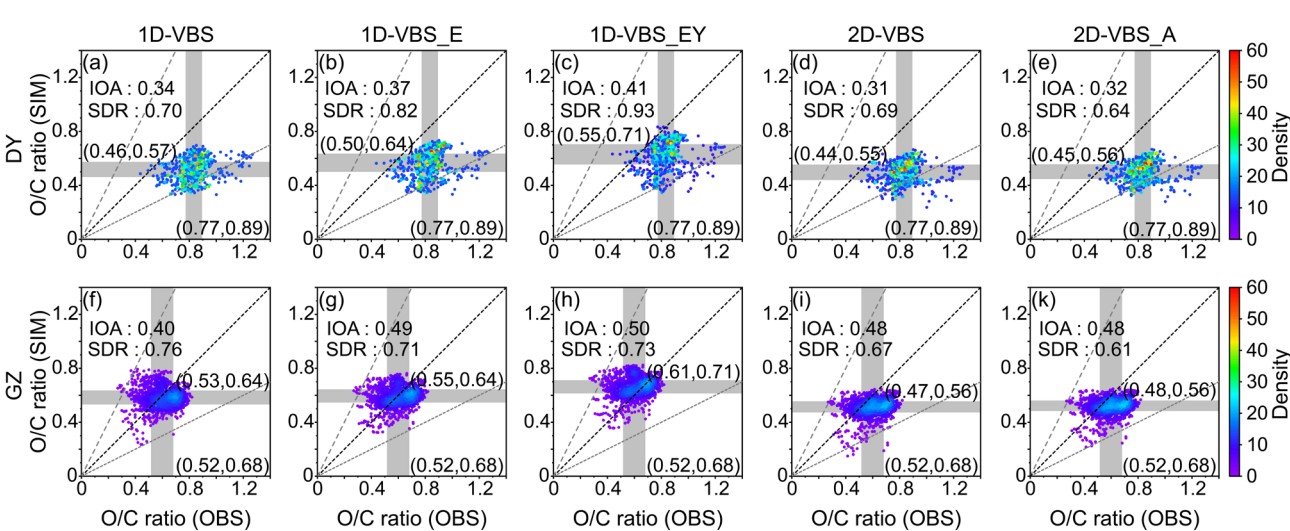

**Figure 4: Density distribution of simulated versus observed hourly O/C ratios of OA in DY (a-e) and GZ (f-k). The edges of the grey shading represent the 25th and 75th percentiles in both the observations and simulations, with the shading indicating the interquartile**

**range (IQR). The Index of Agreement (IOA, reflecting both bias and variability) and Standard Deviation Ratio ($\sigma_{sim}/\sigma_{obs}$, comparing the magnitude of variability in simulations and observations) are calculated.**

The responses of daily OA O/C ratios to ambient $O_3$ concentrations were also examined (Fig. 5). All the simulations

demonstrated an increased in O/C ratios as $O_3$ levels rose (indicating a higher atmospheric oxidative capacity), with the rates



in the range of 0.0023-0.0045 per ppb $O_3$. The 2D-VBS simulations exhibited a steeper slope, suggesting stronger sensitivity

to $O_3$ levels. This implied that oxidant abundance in the gas phase was an important driver of the increase in O/C ratios in the

simulations. However, the observations did not show a clear correlation between O/C ratios and $O_3$. In DY, this could be due

to the decoupling of O/C ratios from local $O_3$ in aged air masses. Other factors, such as fresh emissions in GZ (see Fig. S15)

or aqueous formation pathways, may also contribute to the observed variations in O/C ratios.

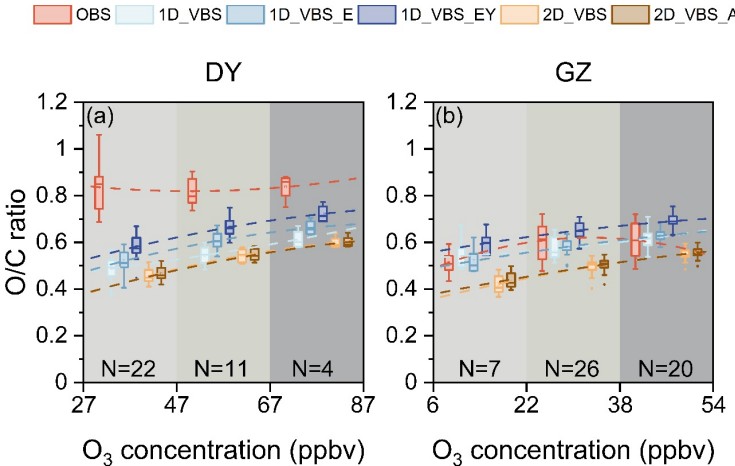

**Figure 5: Simulated changes in OA O/C ratios with daily mean ozone ($O_3$) concentrations, compared to observations in DY (a) and**

**GZ (b). The paired data of O/C ratios and $O_3$ were grouped into three bins with equal intervals of $O_3$ levels. Dashed lines represent**

**fitted trends, and the values of N denote the sample size within each bin.**

**3.4 Implications for Tg and viscosity predictions**

Several widely used CTMs, including WRF-Chem and GEOS-Chem, have been utilized to predict OA viscosity and phase

state based on volatility and O/C (Zhang et al., 2024; Luu et al., 2025). In this study, three parameterizations of $T_g$ were

employed to investigate how model capability in representing volatility and O/C influences the predictions of $T_g$ (and the

resulting viscosity using the VTF equation, see Methods): (1) one incorporating both O/C and volatility (Eq. 2), (2) one

accounting for both volatility and molecular mass (Eq. 3), and (3) one based solely on volatility (Eq. 1). Given that Eq. 3 was

developed from laboratory experiments focused on isoprene SOA, its applicability to POA and ASOA warrants further

evaluation (which is beyond the scope of this study). Nevertheless, it was included here as a comparative parameterization to



evaluate impacts of volatility representations. The simulation at the DY site was selected for detailed analysis as a representative case.

In DY, the mean $T_{g,org}$ values for POA and SOA showed differences across parameterizations, ranging from 15.5 to 24.4 K for POA and 19.3 to 23.0 K for SOA (Fig. 6). The parameterization based on Eq. 1 generally predicted the highest $T_{g,org}$ values, closely aligning with those from Eq.2 as has been reported in Li et al. (2020). This suggests that simulated volatility distributions exerted a stronger influence on $T_{g,org}$ predictions than O/C ratios, as further evidenced by the $T_{g,org}$ values of individual SOA surrogates (Fig. S16). Eq. 3 predicted the lowest $T_{g,org}$ values, including for nearly all SOA components except

a few isoprene SOA surrogates. For isoprene SOA components, $T_g$ estimates remained constant using either Eq. 1 or Eq. 3, due to the use of a fixed $C^*$ of $10^{-10}$ $\mu g\ m^{-3}$ in CMAQ (see Methods), which resulted in unrealistically high $T_g$ values, e.g., ~400 K with Eq.1. However, 2-methyltetrol (AIETET), despite its unexpectedly high ambient concentrations, might be semi-volatile, with predicted $C^*$ of $10^2$ $\mu g\ m^{-3}$, while 2-methyltetrol sulfate (AIEOS) has a relatively low volatility, with $C^*$ of $10^{-1}$ $\mu g\ m^{-3}$ (Budisulistiorini et al., 2017). These values resulted in much lower $T_g$ estimates of approximately 230K and 276K,

respectively (Zhang et al., 2019). Therefore, the $C^*$ values for these compounds should be revisited when applied in $T_g$ predictions.

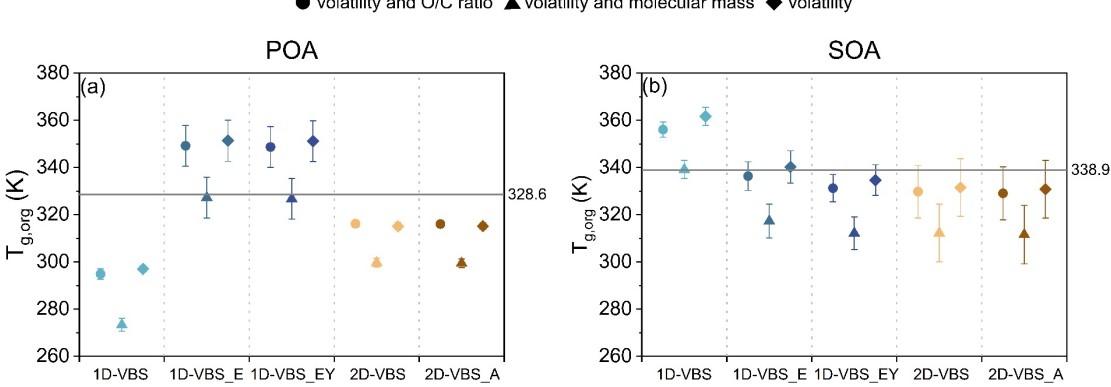

**Figure 6: Predicted $T_{g,org}$ of (a) POA and (b) SOA using three parameterizations across different simulation cases. The solid line represents $T_{g,org}$ derived from observed volatility and O/C ratios. Symbols denote $T_{g,org}$ estimated from: volatility and O/C ratio using**

**Eq. 2 (circles), volatility and molecular weight using Eq. 3 (triangles), and volatility only using Eq. 1 (diamonds). Error bars indicate**





**one standard deviation.**

The discrepancies in POA $T_{g,org}$ derived from observed versus simulated volatility and O/C ratios using Eq. 2 ranged from 12.4 to 33.7 K across the five simulations. Compared to observation-based $T_{g,org}$, the 2D-VBS, 2D-VBS_A, and 1D-VBS

simulations underestimated POA $T_{g,org}$, whereas the 1D-VBS_E and 1D-VBS_EY simulations overpredicted it, primarily due to a substantial contribution from non-volatile POA (Fig. S10). Although the 2D-VBS simulation exhibited the highest fraction of low-volatility POA (Fig. 2), these species were mainly allocated to the volatility bin of $C^*=10^{-2}$ $\mu$g m$^{-3}$, which led to lower $T_{g,org}$ values than those inferred from observations. The biases in SOA $T_{g,org}$ were smaller, ranging from 2.5 to 17.2 K. Most simulations exhibited a slight underestimations of SOA $T_{g,org}$ relative to observation-derived values. Conversely, the 1D-VBS

case overestimated SOA $T_{g,org}$, with an excessive proportion of SOA at $C^*=10^{-10}$ $\mu$g m$^{-3}$ (Fig. S10). Since Eq. 1 and Eq. 2 were developed for use within the 1D-VBS and 2D-VBS frameworks, respectively, the evaluation presented here suggests that the differences in $T_{g,org}$ estimates were driven by the volatility representations in the 1D-VBS versus 2D-VBS, rather than by the specific $T_g$ parameterizations. Notably, the 2D-VBS implementation in CMAQv5.4 generally produced lower SOA $T_{g,org}$ values.

The values of OA viscosity η, which determine their phase state, were calculated using the VTF equation, with $T_g$ parameterized as a function of volatility and O/C ratios (Eq. 2), and incorporating additional variables including $\kappa_{org}$, RH, and T (see Methods). Figure 7a and 7e compared viscosity estimated from simulations and observations, with all input variables (i.e., $T_{g,org}$, $\kappa_{org}$, RH and T) obtained consistently from either model output or observations. In the simulations, $\kappa_{org}$ for each OA species was parameterized as a function of OM/OC (Eq. S7), assuming a constant density of 1.4 g cm$^{-3}$. Bulk $\kappa_{org}$ for SOA and POA was

calculated as a mass-weighted average across species. For observations, $\kappa_{org}$ was inferred from $f_{44}$ (i.e., the fraction of m/z 44 signal in total organic signals) following Feng et al. (2023). The simulations generally reproduced the diurnal variability in OA viscosity, predicting higher η values during the daytime than at night, in line with the observed diurnal patterns. Field measurements indicated that POA was predominantly semi-solid ($10^2 \leq \eta \leq 10^{12}$ Pa·s) with a transition to the solid phase ($\eta > 10^{12}$ Pa·s) between 12:00 and 17:00. However, the model overestimated viscosity. Consequently, the 2D-VBS and 2D-VBS_A

simulations predicted prolonged solid phase for POA, while 1D-VBS_E and 1D-VBS_EY predicted POA to remain solid





throughout the day. For SOA, observations showed that it was semi-solid, while the simulations reproduced this phase state during nighttime hours only.

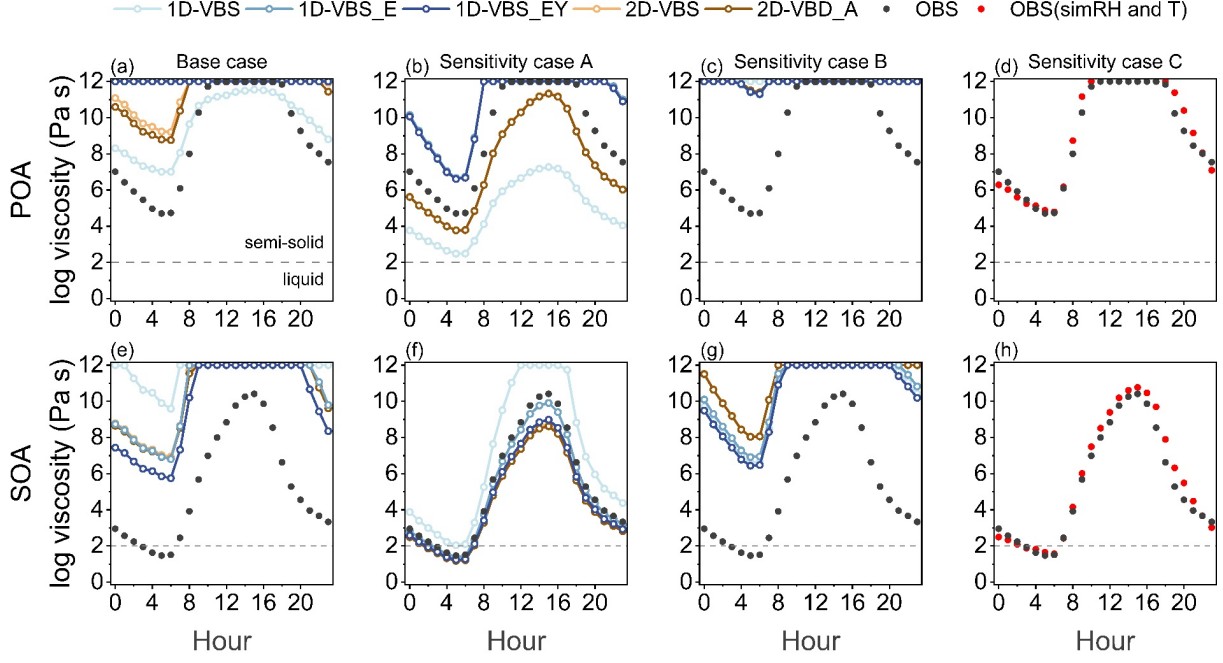

**Figure 7: Diurnal variations in predicted OA viscosity using Eq. 2, based on observations and simulations. Black points indicate observation-based values, while colored lines represent predictions using combinations of $T_{g,org}$, $\kappa_{org}$, RH, and T from observations or simulations as defined in Table 3. Red points indicate viscosity estimated using observation-based $T_{g,org}$ and $\kappa_{org}$ values, along with simulated RH and T. The OA phase state is determined based on viscosity: liquid ($\eta < 10^2$ Pa·s), semi-solid ($10^2 \leq \eta \leq 10^{12}$ Pa·s), and solid ($\eta > 10^{12}$ Pa·s) (Reid et al., 2018). Viscosity values in the figure are capped at $10^{12}$ Pa·s, beyond which values are not physically meaningful for solid-phase OA.**

To identify the dominant source of overestimation in predicted viscosity, three sensitivity experiments were conducted (Table 3), each isolating the influence of a specific input variable. The results from sensitivity experiment A (Fig. 7b and 7f) revealed that the evident model-observation discrepancies in POA and SOA viscosity were positively correlated with the biases in $T_{g,org}$. For instance, in the 1D-VBS case, POA viscosity was significantly underestimated (by up to five orders of magnitude) due to a ~40 K negative bias in $T_{g,org}$ (Fig. S17a), while an overestimation of $T_{g,org}$ for SOA (~20 K, Fig. S17d) led to viscosity



exceeding observation-based values by up to two orders of magnitude. In contrast, simulations such as 1D-VBS_E and 1D-VBS_EY, which better captured SOA $T_{g,org}$, yielded viscosity values in good agreement with observations. Thus, by comparing Fig. 7e and 7f, we conclude that the considerable overestimation of SOA viscosity in Fig. 7e, except in 1D-VBS, cannot be attributed to $T_{g,org}$ bias. O'brien et al. (2021) also reported a moderate sensitivity of predicted OA viscosity to $T_{g,org}$, e.g., under

conditions of RH=60%, a reduction in $T_{g,org}$ by ~200 K resulted in a decrease in predicted viscosity from $10^{12}$ Pa·s to 10 Pa·s (with $\kappa_{org}$ of 0.14).

**Table 3: Combinations of $T_{g,org}$, $\kappa_{org}$, RH and T used in sensitivity cases for OA viscosity estimation.**

| Cases | $T_{g,org}$ | $\kappa_{org}$ | RH and T |
|---|---|---|---|
| Base_case | Simulation-based | Simulation-based | Simulation |
| Sensitivity case A | Simulation-based | Observation-based | Observation |
| Sensitivity case B | Observation-based | Simulation-based | Observation |
| Sensitivity case C | Observation-based | Observation-based | Simulation |

On the other hand, the OA hygroscopicity parameter ($\kappa_{org}$) was an important parameter in viscosity estimation. Sensitivity

experiment B predicted viscosity comparable to those in Fig. 7a and 7e, indicating that the discrepancies between simulation- and observation-based viscosity values was due to the variation in $\kappa_{org}$. Specifically, the $\kappa_{org}$ values used in the simulations underestimated the hygroscopicity of both POA and SOA relative to the values derived from m/z 44 (Fig. S18), e,g, 0.16 versus 0.048-0.069 (1D-VBS)/0.064-0.068 (2D-VBS) for POA, and 0.35 verus 0.013-0.014(1D-VBS)/0.12 (2D-VBS) for SOA. Earlier sensitivity simulations showed that a perturbation of $\pm 0.05$ in $\kappa_{org}$ resulted in changes of $T_{g,\omega org}$ by ~5-15% over high

RH areas, supporting a critical role of $\kappa_{org}$ in viscosity estimation (Shiraiwa et al., 2017). Consistently, the underestimation in $\kappa_{org}$ led to an overestimation in simulated viscosity in this study (given the inverse relationship between viscosity and hygroscopicity), and even offset the underestimations of viscosity that were caused by low biases in $T_{g,org}$ (see Sensitivity experiment A). In CMAQ, $\kappa_{org}$ was parameterized as a function of OM/OC. In light of the notable underestimation of O/C ratios at the DY site (Table 2), this could be a key driver of the bias in viscosity predictions. While the impact of relative



humidity on viscosity has also been emphasized in other studies (Maclean et al., 2021; Rasool et al., 2021; Li et al., 2021b),

its effect is minimal in this study due to the strong agreement between simulated and observed meteorological variables (Fig.

S19). This was further supported by the similarity in viscosity estimates derived using identical $\kappa_{org}$ and $T_{g,org}$ but different RH

and T inputs (Fig. 7d and 7h).

**4 Conclusions and Discussion**

This study evaluated the CMAQ model's performance in predicting OA mass concentrations, volatility distributions, and O/C

ratios, and examined their implications for $T_g$ and viscosity estimates, using observations at two representative sites in Eastern

China, i.e., DY, influenced by aged air masses, and GZ, more impacted by local emissions. The major findings include:

(1)    The base 1D-VBS simulation with AERO7i underestimated OA by ~70%. However, the optimized 1D-VBS simulation,

which incorporated additional L/S/IVOC emissions and enhanced SOA production from aromatics and PAHs (accounting

for vapor wall losses and autooxidation), significantly improved OA predictions with NMBs of −27% (DY).and −34%

(GZ). The 2D-VBS simulation exhibited similar performance with NMBs of −24%. The OA biases were mainly driven

by SOA underestimation, with the optimized 1D-VBS underestimating SOA by ~32% and 2D-VBS underestimating it by

~13%.

(2)    All the simulations accurately captured the observed fraction of low-volatility products in OA (>70%), although the

detailed distributions within these volatility bins ($10^{-9}$ μg m$^{-3}$≤$C^*$≤$0.1$μg m$^{-3}$) were not well represented. Simulated SOA

was abundant in volatility bins of $C^*$=$10^{-2}$ μg m$^{-3}$ or $10^{-10}$ μg m$^{-3}$. The volatility distributions of POA showed notable

variations across the simulations, influenced by the gas/particle partitioning of L/SVOC emissions specified in the model.

(3)    OA O/C ratios were generally underestimated across all the simulations, particularly in DY, where aged air masses

prevailed. In the 1D-VBS simulations, the inclusion of L/S/IVOC emissions increased O/C ratios with more non-volatile

POA, while updated SOA yields added SOA at $C^*$=0.01 μg m$^{-3}$, contributing to higher SOA O/C ratios. The 2D-VBS

simulations presented lower O/C ratios than 1D-VBS overall, whereas constraining POA with observations improved its

O/C representation.

(4) The simulated volatility distribution strongly influenced $T_g$ estimates. In particular, the fraction of POA and SOA in the lowest volatility bin ($C^*=10^{-10}$ µg m⁻³) played an important role. The assumption of $C^*=10^{-10}$ µg m⁻³ resulted in unrealistically high $T_g$ values for some OA components (e.g., isoprene SOA). Although the high (low) biases in $T_g$ led to overestimation (underestimation) of viscosity in the case of DY, uncertainties in OA hygroscopicity parameter, parameterized as a function of O/C ratios in CMAQ, were the dominant source of model-observation gaps in viscosity.

These findings highlight the limitations of the CMAQ model in predicting the volatility distribution of OA and the associated O/C ratios. While the allocation of OA in low-volatility bins may have a limited impact on simulated mass concentrations under atmospherically relevant conditions, it is closely linked to key physicochemical properties, such as $T_g$, viscosity, and phase state. These properties, which have received insufficient attention in previous studies, play a crucial role in the kinetics of multiphase SOA formation. Additionally, estimates of OA hygroscopicity and light absorptivity—both of which influence climate effects—may be subject to uncertainties when parameterized based on O/C ratios. In particular, representing the evolution of O/C ratios during air parcel transport remains a major challenge for CTMs. In some cases, inaccurate treatment of these properties could also influence OA mass concentrations. For example, OA hygroscopicity affects water uptake, which can, in turn, enhance SOA mass through interactions between water and organics. Therefore, future improvements to SOA modeling, including the incorporation of missing precursors, new SOA formation pathways, and more accurate parameterizations, must be rigorously evaluated to ensure they simultaneously reduce model-observation discrepancies in mass concentrations, volatility, and O/C ratios.

**Code and Data availability**

The measurement data are publicly available in previous publications listed in the references. The CMAQ source code can be accessed via the EPA CMAQ GitHub repository (https://github.com/USEPA/CMAQ). CMAQ model output data and Python scripts used to generate the figures are available upon request from the corresponding author.



**Author contributions**

M.Q. and Yu L. designed the research. Yu L. conducted the CMAQ simulations. Yu L. and M.Q. analyzed data, interpreted the results, and wrote the paper, with substantial input from all coauthors.

**Competing interests**

The authors declare that they have no conflict of interest.

**Acknowledgment**

The authors would like to acknowledge the Chinese Ministry of Science and Technology for their support through the Key Research and Development Program (Grant No. 2022YFC3701000). This research was also supported by the National Natural Science Foundation of China (Grant No. 42107117 and 42475124).




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
