# Peer review of "Evaluating Simulations of Organic Aerosol Volatility and Degree of Oxygenation in Eastern China"

_EGUsphere, 2025_

## Author Response (AR1)

We sincerely thank Reviewer #1 for the constructive and insightful comments, which have greatly helped us improve the clarity and quality of our manuscript. We have provided point-by-point responses to each comment. The reviewer's comments are shown in black, and our responses in blue. All revisions to the manuscript are in red. The corresponding changes have been incorporated into the updated manuscript.

Major Comment

**Comment 1-1:**

The authors carefully explored potential reasons for the underestimation of SOA at two sites in China. However, underpredicted POA emissions could significantly affect SOA partitioning and contribute to the observed biases. To evaluate this, I recommend an additional sensitivity simulation in which POA emissions are increased to match observational levels—particularly at the GZ site—to examine whether SOA predictions improve as a result. Furthermore, comparing the diurnal variations of observed and estimated emissions may help identify missing sources and better constrain emission uncertainties. While this would be a sensitivity test, inaccuracies in emission inventories are a well-known issue affecting model performance. In reality, the underprediction of SOA is likely due to a combination of underestimated POA emissions and missing or incomplete SOA formation pathways.

**Response:**

We thank the reviewer for this valuable suggestion. We agree that POA underestimation could lead to reduced SOA partitioning into the particle phase, contributing to low biases. To evaluate this, we conducted sensitivity simulations by scaling POA emissions to better match the observed levels at the two sites. The scaling factors (1.1× at DY and 1.7× at GZ) were calculated as the median of the hourly ratios of observed to simulated POA concentrations over the diurnal cycle. We selected the 1D-VBS_EY case as the baseline because it provided the best overall performance among the simulations. The results are shown in Figure R1-1. The adjusted simulations (green lines) improved the agreement with the observed POA concentrations at both sites. At the GZ site, the increased POA (by 51.6%) also led to higher SOA predictions (by approximately 4.0%) through enhanced gas-particle partitioning, which reduced the bias relative to observations, whereas the improvement at DY was minor. The persistent underestimation of SOA at both sites indicates that the main cause is likely missing or incomplete SOA formation pathways, while the effect of POA underestimation is limited.

[Figure]

**Figure R1-1** Simulated diurnal variations of OA, POA, and SOA at DY (a–c) and GZ (d–f) compared with observations. Blue lines represent the 1D-VBS_EY case, and green lines represent the sensitivity case (1D-VBS_EY_A) with scaled POA emissions. Black dots denote observations. The light grey shading indicates the standard deviations (±1σ) from the mean concentrations in the observations.

We have added this information in the revised manuscript:

Lines 277-281: "Although POA was underestimated (as discussed below), the sensitivity simulations indicated that its impact on SOA was limited (Fig. S8). At the GZ site, for example, increasing POA emissions by 70% in the 1D-VBS_EY simulation to match the observed POA concentrations led to only 4.0% increase in SOA, suggesting that missing or incomplete SOA formation pathways are likely the primary drivers of the SOA low biases."

Since emissions cannot be directly measured or observed, we qualitatively compared the diurnal patterns of observed POA concentrations (which were primarily influenced by both emissions and PBL dynamics) with estimated POA concentrations derived from local emission fluxes and PBL height at the two sites (Figure R1-2). The comparison suggests that the overall magnitude of POA emissions may be underestimated. In addition, the temporal allocation of emissions appears to be overestimated during rush hours but underestimated in the early morning.

[Figure]

**Figure R1-2** Comparison of observed POA concentrations (black dots) with POA concentrations estimated from local emissions and PBL height (purple lines) at DY (a–b) and GZ (c–d). Left panels (a, c) show absolute concentrations, and right panels (b, d) show normalized concentrations relative to the daily median. The shaded areas denote ±1 σ variability of the observations.

Minor Comments

**Comment 1-2:**
"The simulation that updated O/C ratios of POA in 2D-VBS didn't meaningfully affect the predicted mass concentration or volatility distribution … The authors should have a few sentences to explain it."

**Response:**
Thank you for the comment. The modification affects only the chemical composition of POA, without changing their emission magnitudes or volatility parameters ($C^*$). In the 2D-VBS framework, POA mass is largely determined by the emission magnitude and the gas-particle partitioning equilibrium controlled by volatility. Therefore, updating O/C mainly affects predictions of oxidation state or hygroscopicity, while exerting minimal impact on total OA mass or its volatility distributions.

We have revised the manuscript as below:
Lines 254-256 "The simulation with updated O/C ratios of POA emissions in the 2D-VBS framework (i.e., 2D-VBS_A) did not alter the emission magnitudes or volatility, and therefore had a minimal impact on the predicted POA mass concentration or volatility distribution. As a result, this case is not discussed in Section 3.1 and 3.2."

**Comment 1-3:**
"The observations showed that SOA dominated OA at both locations (contributing 72% and 64%, respectively), while the model underestimated this contribution (59%–67%) in DY with the exception of 2D-VBS (82%), and overestimated (72%–84%) it in GZ (Table S9). It reads confusing,

please rephrase it."

**Response:**

We appreciate the reviewer's comment. We have rephrased the sentence as below:

Lines 256–259: "The observations indicated that SOA dominated OA at both sites, contributing 72% and 64% at DY and GZ, respectively. The model generally underestimated the SOA contribution in DY (except for the 2D-VBS case), while slightly overestimating it in GZ (Table S9)."

**Comment 1-4:**

Please add observational points on the figure S7 for comparison.

**Response:**

Thank you for this helpful suggestion. Figure S7 presents the simulated diurnal variations of ASOA and BSOA at the DY and GZ sites. However, the AMS/PMF analysis provides concentrations of LO-OOA and MO-OOA, which cannot be directly mapped to precursor-specific factors, i.e., ASOA and BSOA. Due to the unavailability of corresponding observations, we cannot add the comparison between ASOA/BSOA and LO-OOA/MO-OOA in Figure S7.

**Comment 1-5:**

There are many vague words, like "Modifications" or "Modified", to describe the sensitivity cases. In order to make readers easily understand the objective of cases, the authors would be better to clarify it as "increased", "adding" or "reduced".

**Response:**

Thanks for the suggestion. We have made revisions as below:

Line 95: "Additionally, two adjustments to the default CMAQ scheme were evaluated: (1) adding S/IVOC emissions with SOA formation from these precursors, and (2) updating the SOA yields of aromatic species (benzene, toluene, xylene) and polycyclic aromatic hydrocarbons (PAHs)."

Line 267: "Adding L/S/IVOC emissions …"

**Comment 1-6:**

As CMAQ treats all OA with $C^*$ below $10^{-2}$ µg m$^{-3}$ as non-volatile compounds, to be more comparable with the observations, the authors could merge all the OA with $C^*$ below $10^{-2}$ µg m$^{-3}$ in Figure 3.

**Response:**

We thank the reviewer for the helpful suggestion. We fully agree that merging all OA species with $C^*$ below $10^{-2}$ µg m$^{-3}$ would provide a concise comparison with the observations. The existing Figure S10 in the SI already presents a condensed comparison for low-volatility OA species (C* < $10^{-1}$ µg m-3). However, we intentionally retained Figure 3 in the main text to explicitly show each volatility bin for diagnostic purposes. This detailed representation allows us to examine the modeled differences within the low-volatility and non-volatile ranges relative to observations, which is informative for assessing how the current volatility configuration affects subsequent simulations of

aerosol physicochemical properties (e.g., $T_g$, viscosity). Collapsing these bins into a single category would obscure model biases in volatility and reduce the diagnostic value of the figure. Therefore, we have retained the current figure format in the manuscript.

**Comment 1-7:**

The O/C ratio for GZ and DY sites are quite different, however, it was not captured by any simulations. It might be useful to compare the difference for oxidant concentrations, wind speed, and site environment (e.g., close to highway or residential area) for the observations and model predictions. Although it is difficult to capture this in the model, it is useful to understand why.

**Response:**

We thank the reviewer for this valuable suggestion to further explore the factors contributing to the differences between the simulated and observed O/C ratios.

(1) Site Environment

The two sites differ substantially in their surrounding environments (as also discussed in our response to Comment 2-1 from Reviewer #2), which leads to distinct aging processes. DY is located in the Yellow River Delta National Nature Reserve and strongly influenced by urban outflows from the North China Plain (NCP). On the other hand, the GZ site is in urban GZ surrounded by transportation and residential districts, with significant impacts from local emissions. We attribute the higher observed O/C ratios at DY than at GZ to enhanced chemical aging during regional transport.

While the observed O/C contrast can be explained by these different aging conditions, such aging processes were not adequately represented in the model, resulting in a significant underestimation of the O/C ratio at DY. This conclusion is supported by the simulated O/C differences among the SOA schemes (e.g., 1D-VBS vs. 2D-VBS). For example, the 2D-VBS scheme simulated nearly uniform O/C ratios at both sites and failed to reproduce the observed contrast, indicating that the SOA parameterization is the primary driver of the O/C bias.

(2) Oxidant levels

As suggested, we also compared observed and simulated ozone concentrations at each site (Fig. R1-3). The observations show that DY had a higher mean $O_3$ concentration than GZ (43.0 ppb vs. 28.9 ppb), whereas the two sites exhibited comparable daytime peak $O_3$ levels (119.2 ppb vs. 116.9 ppb). The model generally reproduced the observed $O_3$ concentrations at both sites. These results suggest that differences in oxidant levels are unlikely to be the primary cause of the large O/C contrast observed in the measurements.

[Figure]

**Figure R1-3** Time series of observed and simulated O₃ concentrations at the (a) DY and (b) GZ sites. Black dots represent observations and the red line denotes model simulations. Differences in O₃ among the simulations are negligible.

(3) Meteorology

The wind roses and wind speed distributions indicate that DY experienced stronger and more variable winds, whereas GZ was characterized by weaker, predominantly northerly winds (Figure R1-4). The model partially captured these features but clearly overestimated wind speeds at GZ. Such meteorological biases may influence the simulated O/C ratios. For instance, the excessive dilution caused by overestimated wind speed at GZ would reduce the simulated concentrations of local, freshly formed SOA (which typically has a lower O/C ratio) and increase the relative contribution of SOA transported from upwind (likely with higher O/C). This combined effect would tend to overestimate O/C if the model accurately represented O/C evolution during aging. However, the persistent low bias of O/C at GZ suggests the bias in meteorology is not the main cause compared to the SOA parameterizations.

[Figure]

**Figure R1-4** Observed and simulated wind roses and wind speed probability density functions (PDFs) at the DY (a–d) and GZ (e–h) sites.

We sincerely thank Reviewer #2 for the constructive and insightful comments, which have greatly helped us improve the clarity and quality of our manuscript. We have provided point-by-point responses to each comment. The reviewer's comments are shown in black, and our responses in blue. All revisions to the manuscript are in red. The corresponding changes have been incorporated into the updated manuscript.

Major Comment

**Comment 2-1:**
A detailed comparison of the similarities and differences between the two observation sites (DY and GZ) would strengthen the study. For example, the authors state that DY is influenced by aged air masses while GZ is more impacted by local emissions. However, there is no evidence presented in the study to support this statement. How did the authors come to this conclusion? Additionally, comparing the absolute magnitudes and relative contributions of POA, S/IVOCs, and VOCs from different emission sources at these two sites would help explain the observed and simulated differences between the two sites.

**Response:**
Thank you for this comment. The OA observations used in our study were obtained from two peer-reviewed publications: Feng et al. (2023) for the DY site and Chen et al. (2021) for the GZ site. These two studies provide concrete evidence supporting our statement that *"DY is influenced by aged air masses whereas GZ is more impacted by local emissions."*

Specifically, Feng et al. (2023) identified a substantial "transported OOA" factor that accounted for a major portion (33%) of the observed OA mass concentration at the DY site with AMS/PMF analysis. The elevated transported OOA levels coincided with high concentrations of secondary inorganic aerosols, both of which were driven by southerly winds and by transport from urban areas located to the south of the DY site (e.g., the western and southern regions of Shangdong Province, and the northern regions of Anhui and Jiangsu Provinces), according to the PSCF results derived from backward trajectory analysis. As a result, OA exhibited markedly high oxidation state with O/C = 0.85 (as indicated by $f_{44}$ from the AMS measurements), consistent with extensive aging during long-range transport. In summary, the PMF results, backward trajectory analysis, and high O/C ratios together support the conclusion that DY is influenced by aged air masses.

In contrast, the GZ site was mainly surrounded by urban transportation, commercial and residential districts, as described in Chen et al. (2021). Their PMF results showed significant contributions from hydrocarbon-like OA (HOA) and cooking OA (COA), which accounted for 16% and 8% of total OA, respectively, and exhibited distinct diurnal peaks corresponding to local rush hours and meal times, reflecting strong impacts from local activities. The semi-volatile oxygenated OA (SV-OOA) fraction (27%) peaked at noon (12:00-14:00), indicating that it was freshly produced from local photochemical processes rather than being transported from distant regions. In addition, during polluted periods, static wind speeds and high-pressure systems favor the local accumulation of pollutants.

We have clarified this information in the revised manuscript as follows:

Line 239-241: "One campaign was conducted in the spring (March 17–April 21) of 2018 in DY, at a site located in the Yellow River Delta National Nature Reserve and strongly influenced by urban outflows from the North China Plain (NCP) (Feng et al., 2023)"

Line 246-248: "The other campaign, led by the same team, was conducted in the autumn (September 29-November 21) of 2018 in urban GZ surrounded by transportation and residential districts, with significant impacts from local emissions (Chen et al., 2021)"

We agree that the differences in model performance (observed vs. simulated) are influenced by the different inputs (e.g., emissions) at these two sites. As requested, we performed additional simulations to quantify the absolute magnitudes and relative contributions of POA, S/IVOCs, biogenic VOCs (BVOCs) and anthropogenic VOCs (AVOCs) to total OA (see Figure R2-1 and Table R2-1). This comparative analysis addresses the source differences: (1) The results show that, with the exception of the case 1D-VBS, absolute concentrations of POA and all SOA components were consistently higher at GZ than those at DY across all other simulations. Notably, the simulated OA difference between the two sites was dominated by ASOA and BSOA. (2) The largest contributor to OA at DY was POA across all 1D-VBS simulations, whereas at GZ the dominant contributor varied among simulations. In the 1D-VBS_EY simulation, all precursors contributed almost equally at GZ. However, the 2D-VBS simulation showed S/IVOCs were the largest contributor, followed by AVOCs (with a comparable contribution) at both DY and GZ, indicating SOA parameterization significantly affects source contributions.

[Figure]

**Figure R2-1** Mean concentrations of POA and three SOA components (SOA$_{S/IVOC}$, SOA$_{AVOC}$, and SOA$_{BVOC}$) at (a) DY, and(b) GZ, simulated by the different SOA schemes. SOA$_{S/IVOC}$ represents contributions from semi-volatile/intermediate-volatility organic compounds (S/IVOCs). SOA$_{AVOC}$ includes SOA formed from anthropogenic VOC, predominantly aromatics and alkanes. SOA$_{BVOC}$ represents SOA produced from biogenic VOCs.

Table R2-1 Contribution of OA Components to Total OA in Different Simulations.

| Site | Pollutant | Case | Relative contribution |
|------|-----------|------|-----------------------|
| DY | POA | 1D-VBS | 41% |
| | | 1D-VBS_E | 38% |
| | | 1D-VBS_EY | 33% |
| | | 2D-VBS | 13% |
| | SOA$_{S/IVOC}$ | 1D-VBS | 11% |
| | | 1D-VBS_E | 31% |
| | | 1D-VBS_EY | 27% |
| | | 2D-VBS | 40% |
| | SOA$_{AVOC}$ | 1D-VBS | 18% |
| | | 1D-VBS_E | 12% |
| | | 1D-VBS_EY | 25% |
| | | 2D-VBS | 32% |
| | SOA$_{BVOC}$ | 1D-VBS | 30% |
| | | 1D-VBS_E | 18% |
| | | 1D-VBS_EY | 16% |
| | | 2D-VBS | 15% |
| GZ | POA | 1D-VBS | 22% |
| | | 1D-VBS_E | 28% |
| | | 1D-VBS_EY | 23% |
| | | 2D-VBS | 12% |
| | SOA$_{S/IVOC}$ | 1D-VBS | 6% |
| | | 1D-VBS_E | 27% |
| | | 1D-VBS_EY | 22% |
| | | 2D-VBS | 34% |
| | SOA$_{AVOC}$ | 1D-VBS | 22% |
| | | 1D-VBS_E | 14% |
| | | 1D-VBS_EY | 28% |

| | 2D-VBS | 31% |
|---|---|---|
| | 1D-VBS | 50% |
| | 1D-VBS_E | 31% |
| SOA$_{BVOC}$ | 1D-VBS_EY | 26% |
| | 2D-VBS | 22% |

**Comment 2-2:**

Did the authors include L/S/IVOC emissions from the FINN in their simulations that include L/S/IVOC emissions? Relevant information is not available in Table S3.

**Response:**

Thank you for raising this question. We did not account for L/S/IVOC emissions from wildfire sources in our simulations. In this work, we estimated L/S/IVOC emissions from anthropogenic sources, following the parameterizations used in previous studies. The wildfire emission inventory used in this study—FINN version 1.5, as well as the most recent FINN version 2.5—does not provide explicit L/S/IVOC emission estimates. Therefore, the L/S/IVOC emissions from wildfire were not included in the simulations. Furthermore, the FINN inventory shows that CO emissions from wildfire at both DY and GZ and surrounding aeras were extremely low during the study period, indicating that wildfire emissions were minimal and likely contributed a negligible amount of OA. Consequently, the omission of L/S/IVOC emissions from wildfire sources does not impact the conclusions presented in this study.

**Comment 2-3:**

Model performance evaluation. Why was SO$_2$ not evaluated? The correlation of O$_3$, PM$_{2.5}$, and NO$_2$ is particularly low at the GZ site, and NO$_2$ and PM$_{2.5}$ are substantially underestimated. What are the potential causes for this? Was it due to overestimated wind speed at GZ or bias related to the wind direction?

**Response:**

Thank you for this question. We have added the SO$_2$ performance evaluation to Table S2 in the revised Supplement. The results show that SO$_2$ was underestimated at the GZ site by approximately 40%, suggesting that uncertainties in sulfur emissions may contribute to the underestimation of sulfate and hence to part of the negative PM$_{2.5}$ bias.

While uncertainties in both the emission inventory and simulated meteorological variables may contribute to the low correlations and underestimation of NO$_2$ and PM$_{2.5}$ at the GZ site, we suggest that meteorology, particularly wind speed, likely plays a primary role. As shown in Figure R1-3 in our response to Comment1-7 from Reviewer #1, observed wind speeds at GZ were generally low (typically < 3 m s$^{-1}$), whereas the model substantially overestimated them—a well-known systematic bias in many meteorological models. In the case of GZ, this issue is critical because the polluted periods were associated with stagnant conditions, which favored the

accumulation of pollutants (Chen et al., 2021). The overestimated ventilation diluted locally emitted precursors ($SO_2$, $NO_2$, etc.) and primary $PM_{2.5}$, leading to lower simulated concentrations and reduced temporal correlation with observations. Biases in wind direction were likely less important for model performance at GZ, given that the site was strongly influenced by local emissions and wind direction was more relevant for regionally transported pollutants. In addition, the underestimations of $SO_2$ and $NO_2$ indicate that emissions were likely underestimated, which further contributed to the $PM_{2.5}$ low bias.

**Comment 2-4:**

The authors provided several potential explanations for the underestimated SOA concentrations, such as insufficient formation of organic nitrates and underprediction of aqueous-phase formation pathways. However, POA is also underestimated in all cases. How much would the underestimation of POA contribute to the SOA underestimation, given the gas-particle partitioning relationship? Besides emission uncertainties, could biases in meteorological simulations cause underestimated OA, SOA, and POA?

Response:

Thank you for this insightful comment. To address a similar question raised by Reviewer #1 (Comment 1-1), we conducted a sensitivity simulation in which POA emissions were scaled to match the observed POA concentrations at both the DY and GZ sites. As shown in Figure R1-1, scaling POA to the observed levels increased POA but enhanced SOA by less than 1 µg m$^{-3}$ (up to 4.0%) at the two sites, indicating that the POA underestimation plays only a minor role in the SOA bias.

We agree that meteorological biases may also contribute to the underestimation of OA, POA, and SOA. As stated in our response to Comment 2-3, the model overestimated wind speed under stagnant conditions and exhibited biases in wind direction at both sites. The wind speed bias likely led to excessive dilution of POA, SOA precursors, and SOA, which was likely more important at GZ than at DY. In addition, the wind direction bias may have affected DY more strongly because DY was influenced by regional transport. When the simulated upwind region was cleaner than the actual upwind region, it could have contributed to the low biases in OA at DY site.

**Comment 2-5:**

Table S10. How were the metrics calculated in Table S10? Were they based on hourly pairs of simulation results and observations or were they based on the averaged diurnal data presented in Figure 1?

**Response:**

All statistical metrics in Table S10 were calculated using paired hourly simulation and observation data, rather than the averaged diurnal cycles shown in Figure 1. We have added the term "hourly" to the caption of Table S10 to clarify this.

**Comment 2-6:**

Lines 273-275: The authors mentioned that BBOA and COA were excluded from the observations

when comparing to the simulation results. Did the authors also exclude the simulated POA from biomass burning and cooking? If so, how did the authors exclude simulated POA from these two sources? By performing brute force simulations or using another method? If not, then the POA is even more underestimated.

**Response:**

Thank you for this question. We clarify that we did not exclude the simulated POA from biomass burning and cooking sectors in the model (e.g., via brute force method method). The comparison was made between the "Total simulated POA" and the "Observed POA excluding BBOA and COA."

This adjustment was based on the sectors represented in the emission inventory MEIC and the definitions of PMF factors: (1) The MEIC inventory includes residential biomass burning (e.g., for heating/cooking) but excludes open biomass burning (e.g., in-field burning of agricultural residues and forest/grassland fires) (Li et al., 2019). Although FINN, a global dataset of open biomass burning emissions with a resolution of $0.1° \times 0.1°$ was incorporated into the simulation, it cannot capture tiny fire occurrences. As a result, open biomass burning was likely missing or substantially underestimated in our simulations. The observed BBOA factor, however, includes contributions from both residential and open biomass burning. (2) Additionally, the MEIC inventory primarily accounts for emissions from residential fuel combustion (e.g., coal or gas stoves) but generally lacks emissions from cooking fumes (including oil, ingredients, seasonings, etc.), which the COA factor in observations typically represents. Due to these limitations in the emission inventory, the modeled POA from biomass burning and cooking was highly uncertain.

We agree with the reviewer's assessment. Since the MEIC inventory does include residential biomass burning emissions, comparing the total simulated POA (which includes residential biomass) against the observed POA (with all BBOA removed) creates a mismatch. Ideally, the model should be higher than the adjusted observations. The fact that our simulated POA is still underestimated (despite containing residential biomass emissions that were removed from the observations) suggests that POA emissions are indeed underestimated more significantly than the comparison implies.

**Comment 2-7:**

After comparing mass concentration, O/C ratio, $T_g$, and viscosity, can the authors make specific conclusions or recommendations on parameters or configurations that could improve model performance?

**Response:**

Thank you for this valuable comment. We agree that translating model evaluation results into recommendations is essential for future model development. Based on our comparison of simulated mass concentration, O/C ratio, $T_g$, and viscosity against observations, we draw the following conclusions and recommendations for improving model performance:

(1) Emissions inventory and OA mass concentration

As the simulations suggest significant contributions from L/S/IVOCs to SOA, we recommend refining these emissions, not only in terms of magnitudes but also their volatility distributions. It is also crucial to constrain L/S/IVOCs emissions using measured concentrations of these species in ambient air. In addition, primary emissions from certain sources, such as cooking, open biomass burning, and mobile sources were not well represented and warrant further attention. Lastly, we recommend improving representations of certain SOA formation pathways, particularly via $NO_3$ oxidation and aqueous-phase chemistry, which could enhance night-time SOA mass concentration.

(2) Chemical aging and O/C Ratio

While the 1D-VBS_EY configuration showed improvement by updating SOA yields for aromatics/PAHs, a better approach is to explicitly incorporate autoxidation pathways and the subsequent formation of HOMs, thus improving the modeled O/C ratios. Despite the implementation of multigenerational aging in 1D-VBS_EY and 2D-VBS, the schemes lack constraint from chamber experiments, particularly regarding the O/C evolution with chemical aging. Furthermore, constraining the O/C ratios of POA emissions based on source-specific measurements (e.g., assigning higher O/C values to diesel and gasoline exhaust in this study) is an effective approach for matching the observed POA O/C ratio.

(3) OA $T_g$ and Viscosity

As OA $T_g$ and viscosity predictions are highly sensitive to volatility, it is necessary to revisit the volatility assignments of existing SOA surrogates in the model, such as the isoprene-derived SOA species in CMAQ. Furthermore, developing a more accurate and dynamic parameterization of $\kappa_{org}$, which is currently linked to O/C in CMAQ, is important, as $\kappa_{org}$ also influence viscosity predictions.

Minor Comment

**Comment 2-8:**

There is a reference error in the title of Figure S4. It states "Chen et al. (2024a); (Chen et al., 2024b) and Feng et al. (2023)." However, the legend mentions Chen et al. and Zheng et al.

**Response:**
Revised.

**Comment 2-9:**

Can the authors explain why the L/S/IVOC emissions agree well with Chen et al. (2024) but not Zheng et al. (2023)? Is it due to different methods used?

**Response:**
Thank you for the question. The difference in agreement primarily results from the different methodologies and source data used by the two studies. The L/S/IVOC emissions in our work are consistent with those reported by Chen et al. (2024) because both studies applied the same methodology and data sources. In contrast, Zheng et al. (2023) employed a different approach,

which led to discrepancies relative to our estimates.

We summarize the key differences below:

| | This Work & Chen et al. (2024) | Zheng et al. (2023) |
|---|---|---|
| Methodology | L/S/IVOC emissions were estimated by applying ratios to an existing emission inventory. | Emissions were developed using source- and bin-specific emission factors and activity data. |
| Source Inventory | Based on NMVOC and POA (or POC) emissions from the MEIC inventory (Multi-resolution Emission Inventory for China). | The activity data and removal efficiencies of emission control measures were taken from ABaCAS (Air Benefit and Cost and Attainment Assessment System) emission inventory, while the emission factors and implementation rate of control measures were obtained from Chang et al. (2022). |
| Estimation Details | IVOCs were estimated using source-specific $EF_{IVOC}/EF_{NMVOC}$ ratios, while L/SVOCs were estimated using $EF_{S/LVOC}/EF_{POA(POC)}$ ratios. | Developed using source-specific emission factors for individual volatility bins, combined with detailed activity data obtained from Chang et al. (2022). |

Therefore, the close agreement between our estimates and those reported by Chen et al. (2024) arises from the use of consistent source data (both based on MEIC) and a similar ratio-based methodology. On the other hand, the discrepancy with Zheng et al. (2023) can be attributed to their use of a different primary inventory (ABaCAS) and a different emission-factor–based estimation approach.

We have clarified in the manuscript:
Line 145-150: "The estimated nation-wide L/SVOC and IVOC emissions, which were 3.18 Tg yr$^{-1}$ and 6.68 Tg yr$^{-1}$, respectively, were higher than those reported by Zheng et al. (2023), but agreed well with Chen et al. (2024) in magnitude (Table S4, Figs. S4-S5), as both this study and Chen et al. (2024) applied a ratio-based methodology combined with the MEIC emission inventory. In contrast, Zheng et al. (2023) employed a different approach, using emission factors and activity data obtained from ABaCAS (Air Benefit and Cost and Attainment Assessment System) and Chang et al. (2022), which led to larger discrepancies relative to our estimates."

**Comment 2-10:**
Line 146-147: Please elaborate on "However, the source contributions and volatility distributions of L/S/IVOCs were slightly different."

**Response:**

Thanks for the comment. We have added some interpretations to the manuscript to clarify the specific differences in the estimated L/S/IVOCs:

Line 150-154: "The source contributions and volatility distributions of L/S/IVOCs differed slightly between studies. For example, our estimates indicated a lower contribution of solvent use to IVOCs compared with Zheng et al. (2023) (40% vs. 57%), and a higher contribution from residential sources to S/LVOCs (49% vs. 30%). In addition, the differences in emission magnitudes were primarily in the IVOC volatility range, whereas discrepancies in L/SVOCs were smaller."

**Comment 2-11:**

Figure S13: Add distributions based on observations.

**Response:**

The AMS/PMF data available to us provide only a single average O/C value for each resolved OA factor (e.g., OOA, HOA, COA, BBOA), whereas O/C distributions for ASOA and BSOA are not available.

**Comment 2-12:**

Line 91: CMAQ should be spelled out in the first place mentioned.

**Response:**

Revised.

**References**

Chang, X., Zhao, B., Zheng, H., Wang, S., Cai, S., Guo, F., Gui, P., Huang, G., Wu, D., Han, L., Xing, J., Man, H., Hu, R., Liang, C., Xu, Q., Qiu, X., Ding, D., Liu, K., Han, R., Robinson, A. L., and Donahue, N. M.: Full-volatility emission framework corrects missing and underestimated secondary organic aerosol sources, One Earth, 5, 403-412, https://doi.org/10.1016/j.oneear.2022.03.015, 2022.

Chen, Q., Miao, R., Geng, G., Shrivastava, M., Dao, X., Xu, B., Sun, J., Zhang, X., Liu, M., Tang, G., Tang, Q., Hu, H., Huang, R.-J., Wang, H., Zheng, Y., Qin, Y., Guo, S., Hu, M., and Zhu, T.: Widespread 2013-2020 decreases and reduction challenges of organic aerosol in China, Nat. Commun., 15, 4465, https://doi.org/10.1038/s41467-024-48902-0, 2024.

Chen, W., Ye, Y., Hu, W., Zhou, H., Pan, T., Wang, Y., Song, W., Song, Q., Ye, C., Wang, C., Wang, B., Huang, S., Yuan, B., Zhu, M., Lian, X., Zhang, G., Bi, X., Jiang, F., Liu, J., Canonaco, F., Prevot, A. S. H., Shao, M., and Wang, X.: Real‑Time Characterization of Aerosol Compositions, Sources, and Aging Processes in Guangzhou During PRIDE‑GBA 2018 Campaign, J. Geophys. Res., 126, e2021JD035114, https://doi.org/10.1029/2021JD035114, 2021.

Feng, T., Wang, Y., Hu, W., Zhu, M., Song, W., Chen, W., Sang, Y., Fang, Z., Deng, W., Fang, H., Yu, X., Wu, C., Yuan, B., Huang, S., Shao, M., Huang, X., He, L., Lee, Y. R., Huey, L. G., Canonaco, F., Prevot, A. S. H., and Wang, X.: Impact of aging on the sources, volatility, and viscosity of organic aerosols in Chinese outflows, Atmos. Chem. Phys., 23, 611-636, https://doi.org/10.5194/acp-23-611-2023, 2023.

Li, M., Zhang, Q., Zheng, B., Tong, D., Lei, Y., Liu, F., Hong, C. P., Kang, S. C., Yan, L., Zhang, Y. X., Bo, Y., Su, H., Cheng, Y. F., and He, K. B.: Persistent growth of anthropogenic non-methane volatile organic compound (NMVOC) emissions in China during 1990-2017: drivers, speciation and ozone formation potential, Atmos. Chem. Phys., 19, 8897-8913, 10.5194/acp-19-8897-2019, 2019.

Zheng, H., Chang, X., Wang, S., Li, S., Yin, D., Zhao, B., Huang, G., Huang, L., Jiang, Y., Dong, Z., He, Y., Huang, C., and Xing, J.: Trends of Full-Volatility Organic Emissions in China from 2005 to 2019 and Their Organic Aerosol Formation Potentials, Environ. Sci. Technol. Lett., 10, 137-144, https://doi.org/10.1021/acs.estlett.2c00944, 2023.

---

## Author Response (AR2)

We sincerely thank the editor for the constructive and insightful comments, which have greatly helped us improve the clarity and quality of our manuscript. We have provided point-by-point responses to each comment. The editor's comments are shown in black, and our responses in blue. All revisions to the manuscript are in red. The corresponding changes have been incorporated into the updated manuscript.

Please incorporate comments 1-7 and 2-7 into your manuscript. After doing so, it will be ready for publication.

**Response:**

We thank the editor for the comment. We have incorporated comments 1-7 and 2-7 into the manuscript:

Lines 385-390: "It was also found that wind speeds were overestimated at GZ (Fig. S14), which could lead to excessive dilution and consequently reduce the simulated concentrations of locally emitted POA and freshly formed SOA (both of which typically exhibit lower O/C ratios than aged POA or SOA). This effect would be expected to increase the simulated O/C. However, the low biases in O/C at GZ indicate that meteorological biases are unlikely to be the dominant factor, and that limitations in the SOA representations may play a more important role."

Lines 405-409: "The observations revealed significant differences in the O/C ratios between DY and GZ, reflecting distinct aging processes associated with site characteristics (i.e., regional transport at DY versus dominant local emissions at GZ), rather than differences in oxidant levels, as comparable $O_3$ concentrations were observed at both sites. As discussed above, current SOA parameterizations inadequately represent chemical aging processes and thereby fail to reproduce the observed spatial contrasts in oxidation state."

Lines 553-567: "Therefore, we summarize several recommendations based on this study for future improvements in OA modeling:

(1) Given the substantial contribution of L/S/IVOCs to SOA, emission inventories should be further refined not only in terms of total magnitudes but also volatility-resolved distributions, and constrained using ambient measurements. In addition, primary emissions from sources such as cooking, open biomass burning, and mobile sources require improved representation. Better treatment of nighttime SOA formation pathways, particularly $NO_3$ oxidation and aqueous-phase chemistry, is also needed to reduce SOA mass underestimation.

(2) While updated SOA yields can partially improve model performance, explicitly accounting for autoxidation processes and the formation of HOMs (including both biogenic and anthropogenic origins) would provide a more physically based description of O/C evolution. Moreover, SOA aging schemes should be better constrained by chamber experiments, particularly with respect to the relationship between the degree of oxygenation and multigenerational aging. Constraining POA O/C ratios using source-specific measurements also represents a promising approach for improving the modeled elemental composition of OA.

(3) The linkage between OA volatility and $T_{g,org}$ (and viscosity) requires revisiting the volatility assignments of existing SOA surrogates (e.g., isoprene-derived SOA in CMAQ) and developing more accurate, dynamic parameterizations of $\kappa_{org}$."